# Graph-based Multi-ODE Neural Networks for Spatio-Temporal Traffic Forecasting

**Zibo Liu**[*]                                                                                                      *zbliu@vt.edu*
**Parshin Shojaee**                                                                                    *parshinshojaee@vt.edu*
**Chandan K. Reddy**                                                                                      *reddy@cs.vt.edu*
*Department of Computer Science, Virginia Tech, Arlington, VA*

**Reviewed on OpenReview:** *https://openreview.net/forum?id=Oq5XKRVYpQ*

## Abstract

There is a recent surge in the development of spatio-temporal forecasting models in the transportation domain. Long-range traffic forecasting, however, remains a challenging task due to the intricate and extensive spatio-temporal correlations observed in traffic networks. Current works primarily rely on road networks with graph structures and learn representations using graph neural networks (GNNs), but this approach suffers from over-smoothing problem in deep architectures. To tackle this problem, recent methods introduced the combination of GNNs with residual connections or neural ordinary differential equations (ODE). However, current graph ODE models face two key limitations in feature extraction: (1) they lean towards global temporal patterns, overlooking local patterns that are important for unexpected events; and (2) they lack dynamic semantic edges in their architectural design. In this paper, we propose a novel architecture called Graph-based Multi-ODE Neural Networks (GRAM-ODE) which is designed with multiple connective ODE-GNN modules to learn better representations by capturing different views of complex local and global dynamic spatio-temporal dependencies. We also add some techniques like shared weights and divergence constraints into the intermediate layers of distinct ODE-GNN modules to further improve their communication towards the forecasting task. Our extensive set of experiments conducted on six real-world datasets demonstrate the superior performance of GRAM-ODE compared with state-of-the-art baselines as well as the contribution of different components to the overall performance. The code is available at `https://github.com/zbliu98/GRAM-ODE`

## 1 Introduction

Spatio-temporal forecasting is one of the main research topics studied in the context of temporally varying spatial data which is commonly seen in many real-world applications such as traffic networks, climate networks, urban systems, etc. (Jiang & Luo, 2022; Du et al., 2017; Jones, 2017; Longo et al., 2017). In this paper, we investigate the problem of traffic forecasting, in which the goal is to statistically model and identify historical traffic patterns in conjunction with the underlying road networks to predict the future traffic flow. This task is challenging primarily due to the intricate and extensive spatio-temporal dependencies in traffic networks, also known as intra-dependencies (i.e., temporal correlations within one traffic series) and inter-dependencies (i.e., spatial correlations among correlated traffic series). In addition to this, frequent events such as traffic peaks or accidents lead to the formation of non-stationary time-series among different locations, thus, posing challenges for the prediction task.

Traffic forecasting is a spatio-temporal prediction task that exploits both the location and event data collected by sensors. Traditional methods such as AutoRegressive Integrated Moving Average (ARIMA) and Support Vector Machine (SVM) algorithms only consider the temporal patterns and ignore the corresponding spatial relations (Jeong et al., 2013; Williams & Hoel, 2003; Van Der Voort et al., 1996). Statistical and classical

---

[*]Corresponding author

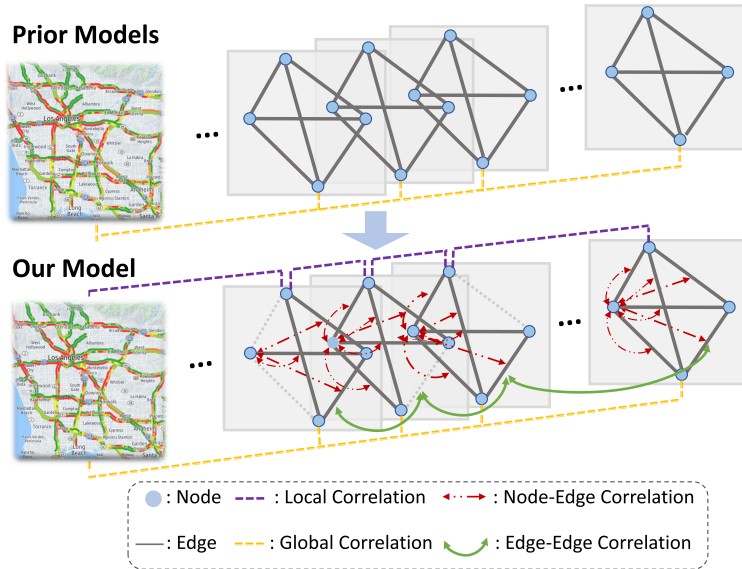

Figure 1: An overview of the proposed model alongside with prior models. The sub-figures depict traffic data over time with blue nodes representing recording points and lines representing roads. The orange dashed line shows the common global temporal pattern. Our model incorporates local temporal patterns represented by the purple dashed line, node-edge correlations depicted by red arrows, and dynamic edge-edge correlations displayed by green arcs.

machine learning methods suffer from limitations in learning complex spatio-temporal interactions, and thus deep learning models were later introduced for this task. Early examples of deep learning approaches to capture spatial and temporal relations are FC-LSTM (Shi et al., 2015) where authors integrate CNN and LSTM modules; and ST-ResNet (Zhang et al., 2017) which uses deep residual CNNs for both spatial and temporal views. However, these CNN-based methods are developed towards grid data and cannnot account for traffic road networks which are more akin to graph-structures. Hence, researchers in the domain have recently developed Graph Neural Network (GNN) based approaches for effectively learning graph-based representations. Examples of these graph-based models are: STGCN (Yu et al., 2018) which utilizes complete convolutional structures for both temporal and spatial views; and DCRNN (Li et al., 2018b) which combines the diffusion process with bi-directional random walks on directed graphs in order to capture spatio-temporal dependencies.

However, such GNN-based methods cannot capture long-range spatio-temporal relations or develop deeper representations due to their limitation of over-smoothing (Lan et al., 2022; Li et al., 2018a; Zhou et al., 2020). Deep GNN over-smoothing occurs when a GNN model with deeper architecture tends to lose the discriminative ability and learn similar node representations for all nodes. Thus, making it challenging to learn richer representations and investigate more complex graph structures. In order to address this problem, researchers have introduced combining GNNs with residual or skip connections (Chen et al., 2020) which are connections that bypass one or more layers and allow information to flow more freely through the network, therefore, improving the network's ability in learning more complex temporal relations. Neural Ordinary Differential Equation (NODE) is a type of deep learning model that uses continuous-time dynamics to learn more powerful representations of the time series data. NODEs can also be used to address over-smoothing in GNNs by providing a more flexible and expressive model architecture in capturing temporal relations. Recently, this was studied by the STGODE (Fang et al., 2021) model that integrates GNNs with NODEs (Chen et al., 2018) to model the dynamics of traffic over time. This combination can derive a continuous GNN (CGNN) with continuous temporal connections toward alleviating the over-smoothing problem. Nevertheless, current CGNN models still encounter the following limitations: (1) Previous approaches in this area tend to overemphasize the global temporal structures while undervaluing local patterns, which are often crucial for predictions in the presence of unexpected traffic events, e.g., a road has a higher chance of getting clogged shortly after a car accident. This will cause drivers to switch to a faster route and thus the traffic flow on this road may significantly drop for a short period of time. Therefore, ignoring these local temporal patterns may cause significant problems in the final predictions for roads facing unexpected events. (2) The dynamic

correlation of traffic nodes is ignored by existing approaches. In other words, models usually do not consider the dynamic semantic spatial correlations (i.e., dynamic semantic edges) in their architecture. (3) Several baselines use vanilla aggregations, such as average pooling, to combine latent features learned from multiple streams which ignore the high-dimensional feature interactions.

To overcome the above limitations, we propose a novel framework called **GRAM-ODE**, **GRA**ph-based **M**ulti-**ODE** Neural Networks. First, in order to balance the consideration of global and local temporal patterns, we design new ODE modules for the local temporal patterns in addition to the existing ODE module for the global temporal patterns (Fang et al., 2021) using different sizes of temporal kernels (represented as purple and orange dashed lines in Fig. 1). Specifically, for local dependencies, we assign ODE functions to the local kernel's output that approximate local patterns, and then concatenate the results. These local and global ODE modules are depicted with more details in Fig. 3(a) and Fig. 3(b), respectively. Second, we design a new ODE module into our model to consider the dynamic correlation of traffic nodes as well as edges. In other words, at each time step, we find the intermediate dynamic spatial correlations based on the embedding representations of nodes (i.e., dynamic semantic edges), and then construct a new ODE module to approximate patterns of semantic edge-to-edge correlations over time (represented with different edge patterns over time in Fig. 1). More details regarding this dynamic semantic edge ODE module are provided in Fig. 3(c). Third, we also design the nonlinear aggregation paradigm and a multi-head attention across different ODE modules (represented in Fig. 3(d)) and different streams of traffic graphs (represented in the last layer of Fig. 2), respectively. Through these two operations, we account for high-dimensional correlations and similarities of latent features corresponding to different ODE modules and traffic graphs. By doing so, we let the model select and fuse latent features from different views for the forecasting task.

Also, since our proposed GRAM-ODE includes multiple ODE modules, we ensure that these different modules are not isolated and have effective connectivity in the intermediate layers. To do so, we design coupled ODE modules in two ways: (1) adding similarity constraint between the semantic parts of local and global modules to ensure these two semantic embeddings do not diverge from one another (represented with red marks in Fig. 3); (2) sharing weights for the global node-based and edge-based ODE modules (represented with green marks in Fig. 3). Therefore, we create a coupled graph-based multi-ODE structure as GRAM-ODE in which all modules are designed to effectively connect with each other for the downstream application of traffic prediction. The major contributions of this work are summarized below.

- *Developing a new ODE module for capturing local temporal patterns.* Due to the importance of short-term temporal dependencies in the traffic prediction in case of unexpected events, we develop a new ODE module for short-term dependencies in addition to the current ODE block for global temporal patterns.

- *Developing a new ODE module for the dynamic semantic edges.* In addition to ODE blocks for traffic nodes, which model the dynamic node-to-node correlations over time, we also add a new ODE module for the dynamic semantic edges based on node representations (i.e., dynamic spatial correlations) to model the edge-to-edge correlations over time.

- *Building effective connections between different ODE modules (coupled multi-ODE blocks).* To build effective interactions between multiple ODE modules, we consider shared weights for node-based and edge-based ODE modules as well as adaptive similarity constraints for the outputs of local and global temporal ODE modules.

- *Designing a new aggregation module with multi-head attention across features of different streams.* To enhance the model's awareness in the selection and fusion of different ODE modules as well as the streams corresponding to different types of traffic graphs, we design multi-head attention mechanism at the aggregation layers.

The rest of this paper is organized as follows. Section 2 summarizes existing traffic forecasting works based on machine learning, graph-based and NODE-based methods. Section 3 provides some of the required preliminaries and explains the problem formulation. Section 4 covers the details of our proposed GRAM-ODE method and its different components. Our experimental evaluation including both quantitative and qualitative comparison results, ablation studies, and robustness assessments are reported in Section 5. Finally, Section 6 concludes the paper.

## 2 Related Works

### 2.1 Machine Learning and Deep Learning Methods

Researchers have employed traditional statistical and machine learning methods for the task of traffic forecasting. Some prominent example models are (1) K-Nearest Neighbor (KNN) (Zhang et al., 2013) which predict traffic of a node based on its k-nearest neighbors; (2) ARIMA (Van Der Voort et al., 1996; Alghamdi et al., 2019) which integrates the autoregressive model with moving average operation; and (3) SARIMA (Williams & Hoel, 2003) which adds a specific ability to ARIMA for the recognition of seasonal patterns. Many of these machine learning models only consider the temporal dependencies and ignore the spatial information. Also, they are usually based on human-designed features and have limitations in learning informative features for the intended task. Later, deep learning methods became popular due to their ability in considering the complex and high-dimensional spatio-temporal dependencies through richer representations. Early examples of deep learning models considering the spatial and temporal relations are FC-LSTM (Shi et al., 2015), ConvLSTM (Xingjian et al., 2015), ST-ResNet (Zhang et al., 2017), ST-LSTM (Liu et al., 2016), and STCNN (He et al., 2019) which are usually based on convolutional neural networks (CNNs) and recurrent neural networks (RNNs) to account for spatial and temporal information. However, these models are developed for the grid-based data and disregard the graph structure of traffic road networks. Due to this limitation, researchers moved into the application of graph-based deep learning models like graph neural networks.

### 2.2 Graph-based Methods

Graphs provide vital knowledge about the spatial, temporal, and spatio-temporal relationships that can potentially improve performance of the final model. Recently, researchers employed the graph convolution network (GCN) (Kipf & Welling, 2017) to model the spatio-temporal interactions. DCRNN (Veeriah et al., 2015) is an early example of it that utilizes diffusion GCN with bi-directional random walks to model the spatial correlations as well as a GRU (Gated Recurrent Unit) based network to model the temporal correlations. GRU is an RNN-based method which is not effective and efficient in modeling long-range temporal dependencies. To address this limitation, works such as STGCN (Yu et al., 2018) and Graph-WaveNet (Wu et al., 2021) utilize convolutional operations for both spatial and temporal domains. After the rise of attention-based models in deep learning, researchers realized that these two models still have some limitations in learning spatial and temporal correlations due to the limited capability of convolutional operations in capturing high-dimensional correlations. Therefore, two attention layers are later employed in ASTGCN (Guo et al., 2019) to capture the spatial and temporal correlations. However, these models are limited in capturing local relations and may lose local information due to the sensitivity of representations to the dilation operation. STSGCN (Song et al., 2020) used localized spatio-temporal subgraphs to enhance prior models in terms of capturing the local correlations. However, since the model formulation does not incorporate global information, this model still had limitations when it came to long-term forecasting and dealing with data that included missing entries or noise. In addition to the spatial graphs from predefined road networks, STFGNN (Li & Zhu, 2021) later introduced the use of Dynamic Time Warping (DTW) for data-driven spatial networks which helped the model to learn representations from different data-driven and domain-driven views. STFGNN also utilized a new fusion module to capture spatial and temporal graphs in parallel. However, due to the over-smoothing problem of deep GNNs (which happens when a GNN model learns similar node representations for all nodes in case of adopting deeper architecture), current GNN-based models incur some difficulties in learning rich spatio-temporal representations with deep layers.

### 2.3 Neural Differential Equation Methods

Neural Differential Equations (NDEs) (Kidger, 2022) provide a new perspective of optimizing neural networks in a continuous manner using differential equations (DEs). In other words, DEs help to create a continuous depth generation mechanism for neural networks, provide a high-capacity function approximation, and offer strong priors on model space. There are a few types of NDEs: (1) neural ordinary differential equation (NODE) such as ODE-based RNN models (Habiba & Pearlmutter, 2020; Lechner & Hasani, 2022) (e.g., ODE-RNN, ODE-GRU, ODE-LSTM) and latent ODE models (Rubanova et al., 2019); (2) neural controlled differential equation (NCDE) (Choi et al., 2022; Kidger et al., 2020; Morrill et al., 2021) which is usually used to learn functions for the analysis of irregular time-series data; and (3) neural stochastic differential equation

Table 1: Notations used in this paper.

| Notation | Description |
|---|---|
| $\mathcal{G}$ | Traffic graph |
| $V$ | Set of nodes |
| $E$ | Set of edges |
| $A^C$ | Connection adjacency matrix |
| $A^{SE}$ | DTW adjacency matrix |
| $D$ | Degree matrix |
| $\hat{A}$ | Normalized adjacency matrix |
| $f_e$/ $f_g$/ $f_l$ | Message generation for Edge/Global/Local Neural ODEs |
| $\mathcal{H}(t)$ | ODE function (integrals from 0 to $t$) initialized with $\mathcal{H}(0)$ |
| $\mathcal{H}_e(t)$/$\mathcal{H}_g(t)$/$\mathcal{H}_l(t)$ | Edge/Global/Local ODE function features |
| $\mathcal{X}$ | Historical time series data |
| $\mathcal{Y}$ | Future time series data |
| $\hat{\mathcal{Y}}$ | Predicted future time series |
| $L$ | Length of historical time series |
| $L'$ | Length of target time series |
| $N$ | Number of nodes, $|V|$ |
| $C$ | Number of channels |
| $H$ | Input of multi ODE-GNN |
| $\mathcal{A}$ | Updated adjacency matrix with shared spatial weights |
| $\mathcal{T}_e$/$\mathcal{T}_g$/$\mathcal{T}_l$ | Updated Global/Local/Edge temporal shared weights |
| **EM** | Edge message |
| **GM** | Global message |
| **LM** | Local message |
| $\mathcal{H}_{Ai}(t)$ | Features of single embedded time step in Local ODE-GNN |
| $L''$ | Length of the embedded temporal window in Local ODE-GNN |
| $\mathcal{K}(i)$ | Local ODE-GNN's output for the $i$-th embedded temporal step |
| $e$ | Learnable threshold parameter in message filtering |
| $p_n$ | Embeddings of $n$-th ODE module in aggregation layer |
| $H'$ | Output of multi ODE-GNN's aggregation layer |
| $W_r$/$b_r$ | Weight matrix/bias vector in update layer |
| $H''$ | Output of multi ODE-GNN's update layer |
| $H^l_{tcn}$ | Hidden states of $l$-th TCN's layer |
| $C^l$ | Embedding size of $l$-th TCN's layer |
| $W^l$ | Convolutional kernel of $l$-th TCN's layer |
| $d^l$ | Exponential dilation rate of $l$-th TCN's layer |
| $W_q$/ $W_k$/ $W_v$ | The attention query/key/value weight matrix |
| $b_q$/ $b_k$/ $b_v$ | The attention query/key/value bias vector |
| $h$ | Number of attention heads |
| $X'_i$ | Attention output of the $i$-th head |
| $\delta$ | Error threshold in the Huber loss |

(NSDE) (Kidger et al., 2021b;a) which is usually employed for generative models that can represent complex stochastic dynamics. More recently, NODEs have been employed for the traffic forecasting task (Fang et al., 2021; Su et al., 2022; Pu et al., 2022). For example, STGODE (Fang et al., 2021) is proposed to address the aforementioned over-smoothing problem of GNN-based models. STGODE provides a new perspective of continuous depth generation by employing Neural Graph ODEs to capture the spatio-temporal dependencies. However, complex spatio-temporal correlations such as the lack of local temporal dependencies and dynamic node-edge communications are not captured by this model. In this study, we employ the idea of Neural Graph ODE for multiple coupled ODE-GNN modules which take into account the local temporal patterns and dynamic spatio-temporal dependencies, and as a result, can provide better function approximation for the forecasting in the presence of complex spatio-temporal correlations.

## 3 Problem Formulation

This section explains the required preliminaries and definitions for GRAM-ODE, and then defines the main problem statement. All notations used in this paper are summarized in Table 1.

### 3.1 Definitions

*Definition 1: Traffic Graphs.* We consider the traffic network as a graph $\mathcal{G} = (V, E, A)$, where $V$ is the set of nodes, $E$ is the set of edges and $A$ is the adjacency matrix such that $A \in \mathbb{R}^{N \times N}$ if $|V| = N$. In this paper, we use two types of graphs, the connection map graph and the DTW (dynamic time warping) graph. In the connection map graph, the adjacency matrix, denoted by $A^C$, represents the roads and actual connectivity among traffic nodes with binary value.

$$A^C_{i,j} = \begin{cases} 1, & \text{if } v_i \text{ and } v_j \text{ are neighbors} \\ 0, & \text{otherwise} \end{cases} \tag{1}$$

where $v_i$ and $v_j$ refer to traffic nodes $i$ and $j$ in the graph. In the DTW graph, the adjacency matrix, indicated by $A^{SE}$, is generated from the Dynamic Time Warping (DTW) algorithm (Berndt & Clifford, 1994) which calculates the distance between two time-series corresponding to a pair of nodes.

$$A_{i,j}^{SE} = \left\{ \begin{array}{l} 1, DTW\left(X^i, X^j\right) < \epsilon \\ 0, \text{ otherwise} \end{array} \right. \tag{2}$$

where $X^i$ and $X^j$ refers to the time-series data of nodes $i$ and $j$, respectively; $\epsilon$ also identifies the sparsity ratio of adjacency matrix. Notably, DTW is more effective than other point-wise similarity metrics (such as Euclidean distance) due to its sensitivity to shape and pattern similarities.

*Definition 2: Graph Normalization.* Adjacency matrix $A \in \{A^C, A^{SE}\} \in \mathbb{R}^{N \times N}$ is normalized with $D^{-\frac{1}{2}}AD^{-\frac{1}{2}}$ where $D$ is the degree matrix of $A$. As shown in Eq. (3), the self-loop identity matrix $I$ is incorporated in normalization to avoid the negative eigenvalues.

$$\hat{A} = \alpha \left( I + D^{-\frac{1}{2}}AD^{-\frac{1}{2}} \right) \tag{3}$$

where $\hat{A}$ is the normalized adjacency matrix; and $\alpha \in (0, 1)$ identifies the eigenvalue range to be in $[0, \alpha]$.

*Definition 3: Graph-based Neural ODE.* The standard formulation of GNN-based continuous-time ODE function is defined in Eq. (4). It takes the initial value $\mathcal{H}(0)$, temporal integrals from 0 to given time $t$, traffic graph $\mathcal{G}$, and the Neural ODE's network parameter $\theta$.

$$\mathcal{H}(t) = \mathcal{H}(0) + \int_0^t f(\mathcal{H}(s), s; \mathcal{G}, \theta)\mathrm{d}s \tag{4}$$

where $f$ is the process to generate semantic message of Neural ODE parameterized by $\theta$ to model the hidden dynamics of graph $\mathcal{G}$. Since the structure of our proposed method consists of multiple ODE-GNN blocks, we have different types of Eq. (4) graph neural ODEs which are explained with more details in Section 4.

*Definition 4: Tensor n-mode multiplication.* We use subscript to identify the tensor-matrix multiplication on the specific dimension as shown below.

$$(\mathcal{B} \times_2 \mathcal{C})_{ilk} = \sum_{j=1}^{n_2} \mathcal{B}_{ijk}\mathcal{C}_{jl} \tag{5}$$

where $\mathcal{B} \in \mathbb{R}^{N_1 \times N_2 \times N_3}$, $\mathcal{C} \in \mathbb{R}^{N_2 \times N_2'}$, so, $\mathcal{B} \times_2 \mathcal{C} \in \mathbb{R}^{N_1 \times N_2' \times N_3}$. The $n$-mode tensor-matrix multiplication is denoted as $\times_n$ with the $n$-th subscript.

### 3.2 Problem Statement

The spatio-temporal forecasting problem is described as learning a mapping function $\mathcal{F}$ that transforms the current historical spatio-temporal data $\mathcal{X} = (X_{t-L+1}, X_{t-L+2}, ..., X_t)$ into the future spatio-temporal data $\mathcal{Y} = (X_{t+1}, X_{t+2}, ..., X_{t+L'})$, where $L$ and $L'$ denote the length of historical and target time-series to be predicted, respectively. In the traffic forecasting problem, we have a historical tensor $\mathcal{X} \in \mathbb{R}^{B \times N \times L \times C}$ and traffic graph $\mathcal{G}$, where $B$ is the batch size; $N$ is the number of nodes; $L$ is the input temporal length; and $C$ is the number of input features (e.g., traffic speed, flow, density). The goal is to find $\hat{\mathcal{Y}} = \mathcal{F}(\mathcal{X}, f, \mathcal{G})$ in which $\mathcal{F}$ is the overall forecasting network and $f$ corresponds to different graph-based Neural ODE processes.

## 4 GRAM-ODE

### 4.1 Overview

The overview of our proposed GRAM-ODE is shown in Fig. 2, which is composed of two streams of operations: (*i*) DTW-based graph (top row) which is constructed based on semantical similarities, and (*ii*) connection map graph (bottom row) which is constructed based on geographical spatial connectivities. These two types of adjacency matrices are fed into the model separately to capture spatial correlations from both data-driven and geographical domain-driven views. They are later integrated with the multi-head attention mechanism in the final layer. As shown in Fig. 2, we have three parallel channels for each graph, each of which has two consecutive GRAM-ODE layers with a multi ODE-GNN block sandwiched between two blocks of temporal

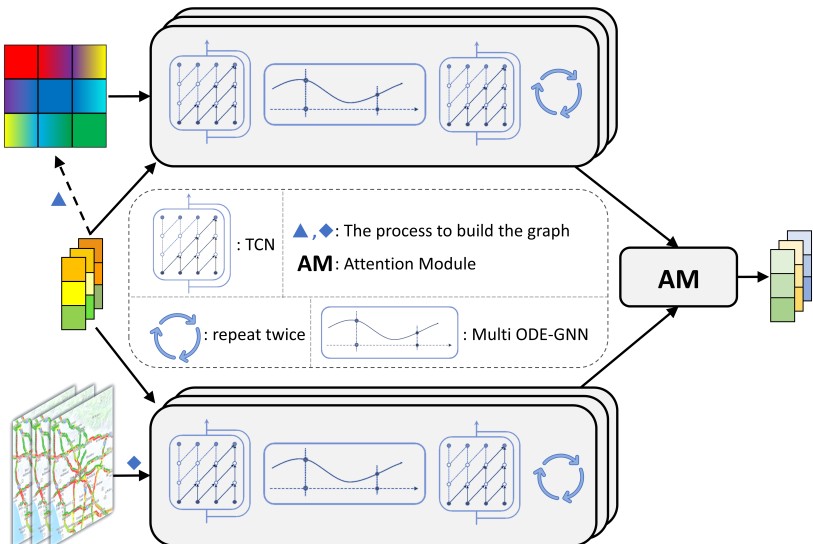

Figure 2: A graphical overview of the proposed GRAM-ODE framework. The DTW-based graph will be obtained from data at the blue triangle mark. The connection map will be obtained from the distance among recording nodes at the blue diamond mark. These two traffic graphs are separately imported into the model which consists of three parallel channels of two consecutive GRAM-ODE layers. Each layer contains a sandwiched multi ODE-GNN block (explained in Fig. 3) between two temporal convolution networks (TCNs). Outputs of these two streams are then aggregated with an Attention Module ($AM$) in the final layer.

dilated convolution (TCN). The final features of each graph from different channels are then concatenated and imported into the attention module ($AM$) which is designed to effectively fuse these two separate sets of operations by taking into account all the high-dimensional relations towards the intended forecasting task. We provide further details about each of these components in the subsections below.

## 4.2 GRAM-ODE Layer

Each GRAM-ODE layer consists of a multi ODE-GNN block placed between two TCNs. Fig. 3 illustrates details of our proposed Multi ODE-GNN block in which we combine multiple ODE modules to take into account all the essential temporal and spatial patterns from different viewpoints and extract informative features from their fusion. In the multi ODE-GNN block, we design three types of message passing operations, a message filtering constraint as well as a new aggregation module to consider high-dimensional correlations.

### 4.2.1 Message Passing Layer

The current models primarily focus on node-based global temporal correlations and overlook short-term temporal patterns as well as the dynamic edge-based correlations in their formulation. Hence, we design three types of message passing processes to formulate ODE modules corresponding to global, local, and edge-based temporal dependencies.

*Shared Weights:* Although node-based and edge-based features have distinct semantic meanings, they can still exhibit common spatial and temporal patterns. In order to consider these common patterns in our problem formulation, we design a shared weight matrix between node-based and edge-based global temporal ODE modules (shown in Fig. 3(a) and 3(c)). We define two weight matrices for consideration of these shared spatial and temporal weights. The shared spatial weight, denoted by $M$, is added to the normalized adjacency matrix as $\hat{A} = \hat{A} + M$, where $M$ is initialized randomly from the normal distribution. The shared temporal weights, denoted by $W_{s1}, W_{s2}$, is added to the node-based and edge-based global temporal modules, given in Eqs. (6) and (7). To consider the local temporal patterns in model formulation, we apply different sizes of temporal kernels $W_{l1}, W_{l2}$ into the local temporal module as shown in Eq. (8).

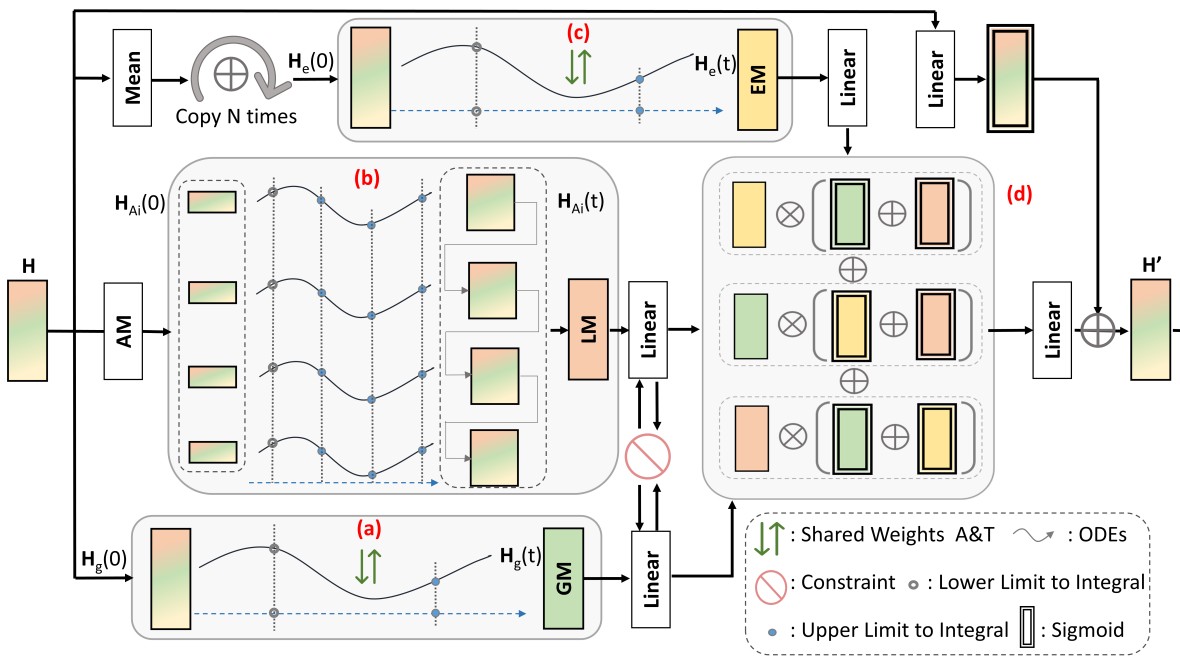

Figure 3: An overview of the multi ODE-GNN block which consists of three ODE modules, i.e., (a) global, (b) local, and (c) edge-based temporal dependencies as well as a (d) new aggregation layer. The inputs and outputs of the multi ODE-GNN block are displayed with $H$ and $H'$ blocks on the left and right sides of the diagram. The shared weights among different ODE modules are marked in green, and a constraint to limit the divergence of embeddings is marked in red. AM denotes the Attention Module defined in Section 4.3.

$$\mathcal{T}_e = (\mathcal{H}_e(t) \times_4 W_{s1})^T (\mathcal{H}_e(t) \times_4 W_{s2}) \tag{6}$$

$$\mathcal{T}_g = (\mathcal{H}_g^T(t) \times_3 W_{s1})(\mathcal{H}_g^T(t) \times_3 W_{s2})^T \tag{7}$$

$$\mathcal{T}_l = (\mathcal{H}_l^T(t) \times_3 W_{l1})(\mathcal{H}_l^T(t) \times_3 W_{l2})^T \tag{8}$$

where $W_{s1}, W_{s2} \in \mathbb{R}^{L \times L}$ represent the global temporal shared weights with $L$ referring to the global temporal length; $W_{l1}, W_{l2} \in \mathbb{R}^{1 \times 1}$ represent the local temporal weights for each time step in the embedded temporal space; and $\mathcal{H}_e(t), \mathcal{H}_g(t)$, and $\mathcal{H}_l(t)$ represent the feature of edge, global, and local ODE module, respectively.

*Global and Local Message Passing:* Fig 3(a) represents the global message passing with the goal of modeling long-term node-based temporal patterns in a continuous manner. Eqs. (9) and (10) represent the global message processing operations and return the global message **GM**, respectively. Additionally, Fig 3(b) represents the local message passing operations on local temporal data to emphasize the importance of short-term temporal patterns. In this module, we first use self-attention mechanism with dense projection layers to create the input in the lower-dimensional embedded temporal space by considering all the possible high-dimensional correlations (similar to the attention module ($AM$) explained with more details in Section 4.3). Then, features of each time stamp in the embedded temporal inputs are separately imported into the local ODE functions, encouraging the network to learn implicit local dependencies. These features at each embedded time step are denoted by $\mathcal{H}_{Ai} \in \mathbb{R}^{B \times N \times 1 \times C}$, where $i \in \{0, 1, 2, 3, ..., L'' - 1\}$, and $L''$ is the size of the embedded temporal window. As shown in Eqs. (13) and (15), at each embedded time step, $\mathcal{H}_{Ai}$ is used as the initial value to formulate temporal dependencies of future $t = L/L''$ time stamps which are returned as $\mathcal{K}(i)$. Then, the outputs are concatenated to form the final local message **LM** $= \mathcal{K}(0) || \mathcal{K}(1) || \ldots || \mathcal{K}(L'' - 1)$.

$$\mathbf{GM} = GlobalMessagePassing(\mathcal{H}_g(0), \hat{\mathcal{A}}, \mathcal{T}_g, \mathcal{W}) = \mathcal{H}_g(0) + \int_0^t f_g(\mathcal{H}_g(\tau), \hat{\mathcal{A}}, \mathcal{T}_g, \mathcal{W}) \mathrm{d}\tau \tag{9}$$

$$f_g = \mathcal{H}_g(t) \times_2 (\hat{\mathcal{A}} - I) + ((S(\mathcal{T}_g) - I)\mathcal{H}_g^T(t))^T + \mathcal{H}_g(t) \times_4 (\mathcal{W} - I) \tag{10}$$

$$\mathbf{LM} = LocalMessagePassing(ATT, \mathcal{H}_l(0), \hat{\mathcal{A}}, \mathcal{T}_l, \mathcal{W}) = \mathcal{K}(0)||\mathcal{K}(1)||......||\mathcal{K}(L''-1) \tag{11}$$

$$\mathcal{H}_{A0}, \mathcal{H}_{A1}, ..., \mathcal{H}_{Al} = ATT(\mathcal{H}_l(0)) \tag{12}$$

$$\mathcal{K}(i) = F_l(\mathcal{H}_{Ai}, t_0)||F_l(\mathcal{H}_{Ai}, t_1)||......||F_l(\mathcal{H}_{Ai}, t_{L/L''-1}) \tag{13}$$

$$F_l(\mathcal{H}_{Ai}, t_j) = \mathcal{H}_{Ai} + \int_0^{t_j} f_l(\mathcal{H}_{Ai}(\tau), \hat{\mathcal{A}}, \mathcal{T}_l, \mathcal{W})\mathrm{d}\tau \tag{14}$$

$$f_l = \mathcal{H}_{Ai}(t) \times_2 (\hat{\mathcal{A}} - I) + ((S(\mathcal{T}_l) - I)\mathcal{H}_{Ai}^T(t))^T + \mathcal{H}_{Ai}(t) \times_4 (\mathcal{W} - I) \tag{15}$$

*Edge Message Passing:* Fig. 3(c) depicts the edge message passing procedures that take into account the dynamic edge-based temporal patterns in addition to the node-based correlations. We first create the initial edge features $H_e(0) \in \mathbb{R}^{B \times N \times N \times L}$ from the node representation $H \in \mathbb{R}^{B \times N \times L \times C}$ by taking an average over $C$ channels and copying the $N$ dimension. Eqs. (16) and (17) represent the edge message passing operations which return edge message **EM**.

$$\mathbf{EM} = EdgeMessagePassing(\mathcal{H}_e(0), \hat{\mathcal{A}}, \mathcal{T}_e) = \mathcal{H}_e(0) + \int_0^t f_e(\mathcal{H}_e(\tau), \hat{\mathcal{A}}, \mathcal{T}_e)\mathrm{d}\tau \tag{16}$$

$$f_e = \mathcal{H}_e(t) \times_2 (\hat{\mathcal{A}} - I) + \mathcal{H}_e(t)(S(\mathcal{T}_e) - I) \tag{17}$$

In Eqs (9)-(17), $\mathcal{H}_g$, $\mathcal{H}_l$, and $\mathcal{H}_e$ represent the feature of global, local, and edge ODE modules, repsectively; $\mathcal{T}_g$, $\mathcal{T}_l$, and $\mathcal{T}_e$ represent the global, local, and edge temporal weights formulated in Eqs. (6)-(8), respectively; $\hat{\mathcal{A}}$ represents the updated adjacency matrix based on shared spatial weights; $\mathcal{W} \in \mathbb{R}^{C \times C}$ represents the weight matrix for modeling interaction of different channels; $S(.)$ shows the sigmoid operation; and $I$ is the identity matrix.

### 4.2.2 Message Filtering

To prevent the semantic parts of local and global modules diverging from one another, we add a message filtering constraint into the problem formulation (shown with red mark in Fig. 3). Eq. (18) shows that this message filtering constraint works like a both-way clipping operation in which we clip local semantic embeddings if they are significantly larger or smaller than global semantic embeddings.

$$LM = \begin{cases} GM + e, \, if \, \, LM > GM + e \\ GM - e, \, if \, \, LM < GM - e \\ LM, \, otherwise \end{cases} \tag{18}$$

where $LM$ and $GM$ represent the local and global semantic embeddings; and $e$ represents a learnable noise parameter initialized with normal distribution.

### 4.2.3 Aggregation Layer

Fig. 3(d) represents our aggregation paradigm designed for combining output features of the three ODE modules. Instead of employing single add or pooling operations in the aggregation layer, we use the non-linear matrix multiplication operation to account for key correlations in the higher dimensions. Eq. (19) represents our designed aggregation in which the output of each ODE module is multiplied by the sum of other modules that are normalized by softmax operation. This gated aggregation has several benefits: (*i*) it enables the selection of features that are more crucial for forecasting; (*ii*) it allows for non-linear aggregation; and (*iii*) it contains a linear gradient path, reducing the risk of vanishing gradient.

$$H' = Aggregation(GM, LM, EM) = \frac{1}{2K} \sum_m^K \sum_{n \neq m}^K p_m \odot softmax(p_n) \tag{19}$$

where $H'$ represents the output of Multi ODE-GNN aggregation layer; $LM$, $GM$, and $EM$ represent the local, global, and edge-based semantic embeddings, respectively; $K = 3$ refers to three ODE modules ($p_0 = GM$, $p_1 = LM$, $p_2 = EM$); and $\odot$ operation represents the point-wise multiplication.

### 4.2.4 Update Layer

After obtaining the aggregated information, we add a residual connection around the multi ODE-GNN block to update the node information of GNN. To achieve this, the input of multi ODE-GNN block is remapped to the aggregated output using a Fully Connected Layer as shown below.

$$H'' = Update(H', H) = \alpha * Sigmoid(W_r H + b_r) + \beta * H' \tag{20}$$

where $H$ and $H'$ represent the inputs and outputs of the multi ODE-GNN block, respectively; $W_r$ and $b_r$ denote the weight matrix and bias vector of the residual fully connected layer, respectively. Also, $\alpha$ and $\beta$ are the hyperparameters identifying the combination weights in the residual connection.

### 4.2.5 Temporal Convolutional Network

Since the problem being considered in this paper is spatio-temporal in nature, we need to consider the temporal correlations of nodes in addition to their spatial relations in the problem formulation. To model the temporal correlations, we utilize temporal convolutional networks (TCNs) which usually have more efficient training, more stable gradients, and faster response to dynamic changes compared to recurrent neural networks (RNNs). The architecture of TCN adopts the dilated convolutional operations with 1-D kernels along the time axis.

$$H_{tcn}^l = \begin{cases} X & , \quad l = 0 \\ Sigmoid\left(W^l \odot d^l H_{tcn}^{l-1}\right) & , \quad l = 1, 2, \ldots, L \end{cases} \tag{21}$$

where $X \in \mathbb{R}^{B \times N \times L \times C}$ is the input of TCN; $H_{tcn}^l \in \mathbb{R}^{B \times N \times L \times C^l}$ is the latent output with $C^l$ as the embedding size in the $l$-th TCN's layer; $W^l$ represents the convolutional kernel of the $l$-th layer; and $d^l$ is the exponential dilation rate which is used to expand the receptive field during the convolution. We usually take $d^l = 2^{l-1}$ in the temporal dilated convolution.

Algorithm 1 provides the pseudocode for the GRAM-ODE layer by sequentially passing the input via TCN, Multi ODE-GNN, and another TCN blocks. The previously explained steps of Multi ODE-GNN block including initialization, message passing, message filtering, aggregation, and update are summarized in lines 4 - 26.

---

**Algorithm 1:** GRAM-ODE Layer

**Input:** Node Information $\mathcal{X}$, Traffic Graph $\mathcal{G}$
**Output:** Updated Node Information $\hat{\mathcal{X}}$

1   # TCN Block
2   $H \leftarrow \text{TCN}(\mathcal{X})$
3   # Find Initial Values
4   $\mathcal{H}_g(0) \leftarrow H$
5   $\mathcal{H}_{Ai}(0) \leftarrow \text{ATT}(H)$
6   $\hat{H} \leftarrow \text{Mean}(H)$ # average on channel dimension
7   $H_e(0) \leftarrow \text{Repeat}(\hat{H})$ # repeat N times
8   # Edge Message Passing
9   $EM \leftarrow \mathcal{H}_e(t) \times_2 (\hat{A} - I) + \mathcal{H}_e(t)(S(\mathcal{T}_e) - I)$
10   # Global Message Passing
11   $GM \leftarrow \mathcal{H}_g(t) \times_2 (\hat{A} - I) + ((S(\mathcal{T}_g) - I)\mathcal{H}_g^T(t))^T + \mathcal{H}_g(t) \times_4 (\mathcal{W} - I)$
12   # Local Message Passing
13   $LM \leftarrow \mathcal{H}_{Ai}(t) \times_2 (\hat{A} - I) + ((S(\mathcal{T}_l) - I)\mathcal{H}_{Ai}^T(t))^T + \mathcal{H}_{Ai}(t) \times_4 (\mathcal{W} - I)$
14   # Message Filtering
15   **if** $LM > GM + e$ **then**
16     |   $LM \leftarrow GM + e$
17   **else if** $LM < GM - e$ **then**
18     |   $LM \leftarrow GM - e$
19   **else**
20     |   $LM \leftarrow LM$
21   **end**
22   # Aggregate Multi ODE-GNNs
23   $p_0, p_1, p_2 \leftarrow GM, LM, EM$
24   $H' \leftarrow \frac{1}{2K} \sum\limits_{m}^{K} \sum\limits_{n \neq m}^{K} p_m \odot softmax(p_n)$
25   # Update
26   $H'' \leftarrow \alpha * Sigmoid(W_r H + b_r) + \beta * H'$
27   # TCN Block
28   $\hat{\mathcal{X}} \leftarrow \text{TCN}(H'')$

---

### 4.3 Attention Module

We use the attention mechanism in the last layer of GRAM-ODE to effectively aggregate the final learned embeddings of two traffic graphs (i.e., DTW-based graph and the connection map graph) in a way that is better aligned with the forecasting objective. The attention module (AM) is designed to replace the previous fully connected layers while capturing the correlations of high-dimensional information. In this module, we first concatenate the embeddings of two graphs and then compute the attention scores between them. Eqs. (22) and (23) mathematically describe this attention operation.

$$Q = XW_q + b_q, \ K = XW_k + b_k, \ V = XW_v + b_v \tag{22}$$

$$X'_i = softmax\left(\sqrt{\frac{h}{C'}} * (Q_i^T K_i)\right) V_i \tag{23}$$

where $W_q(b_q), W_k(b_k)$, and $W_v(b_v)$ represent the attention query, key, and value weight matrix (bias vector), respectively; $X$ is the input of attention module; $h$ is the number of heads; and $\sqrt{\frac{h}{C'}}$ is the normalization factor with $C'$ representing the embedding dimension. Therefore, $Q:\{Q_1, Q_2, ..., Q_h\}$, $K:\{K_1, K_2, ..., K_h\}$, $V:\{V_1, V_2, ..., V_h\}$ shows the query, key, and value set for multiple attention heads. Finally, output of the attention module $X'_i$ can be mapped to the feature space of original data for prediction with a linear dense layer.

### 4.4 Loss Function

In regression problem settings (such as the traffic flow forecasting), Huber loss function between the real value and predicted value is widely used in the literature. It combines the merits of $L1$ and $L2$ loss functions. Huber loss is a piece-wise function which consists of two parts: (1) a squared term with small and smooth gradient when the difference between the true and predicted values is small (i.e., less than a $\delta$ threshold value) and (2) a restricted gradient term when the true and predicted values are far from each other. Eq. (24) represents the standard form of Huber loss function.

$$L(\hat{\mathcal{Y}}, \mathcal{Y}) = \begin{cases} \frac{1}{2}(\hat{\mathcal{Y}} - \mathcal{Y})^2, & |\hat{\mathcal{Y}} - \mathcal{Y}| \leq \delta \\ \delta|\hat{\mathcal{Y}} - \mathcal{Y}| - \frac{1}{2}\delta^2, & otherwise \end{cases} \tag{24}$$

where $\delta$ is a hyperparameter value set for the intended threshold; $\mathcal{Y}$ is the true future spatio-temporal data; and $\hat{\mathcal{Y}}$ is the predicted future data.

Algorithm 2 provides the complete pseudocode of GRAM-ODE training. In each optimization step of this algorithm, the outputs of all GRAM-ODE layers across parallel channels are concatenated, and these concatenated outputs of different graph types are then fed into the attention module for better aggregation and prediction.

---

**Algorithm 2:** GRAM-ODE training

**Input:** Historical Data $\mathcal{X}$, Future Data $\mathcal{Y}$, Traffic Graph $\mathcal{G}$
**Output:** Forecast Model with parameter $\theta$

1   initialize model parameters $\theta$
2   normalize the historical data $X \leftarrow \frac{\mathcal{X} - mean(\mathcal{X})}{std(\mathcal{X})}$
3   **for** *number of epochs until convergence* **do**
4     **for** *batch in num_batches* **do**
5       layers=[ ]
6       **for** *graph g in $\mathcal{G}$* **do**
7         **repeat**
8           $\hat{X} \leftarrow$ GRAM-ODE Layer$(X, g)$
9           layers $\leftarrow$[layers, $(\hat{X})$]  # Concatenation
10         **until** *num_paralleled_layers*
11       **end**
12       $X' \leftarrow$ Attention Module(layers)
13       $\hat{\mathcal{Y}} \leftarrow X' \times std(\mathcal{X}) + mean(\mathcal{X})$
14       $l \leftarrow$ Huber Loss$(\hat{\mathcal{Y}}, \mathcal{Y})$ # Compute Loss
15       $\theta \leftarrow \theta - \Delta_\theta l$ # Update Parameters
16     **end**
17   **end**

---

## 5 Our Experiments

We conduct experiments on six real-world datasets and seven baseline models to evaluate the effectiveness of our proposed GRAM-ODE and its components for the traffic forecasting task.

### 5.1 Datasets

We show the performance results of our model on six widely used public benchmark traffic datasets [1] : PEMS03, PEMS04, PEMS07, and PEMS08 released by (Song et al., 2020) as well as PEMS-BAY (Li et al., 2017) and METR-LA (Jagadish et al., 2014). The first four datasets (PEMS03, PEMS04, PEMS07, PEMS08) are collected based on the California Districts they represent (District 3, District 4, District 7, and District 8, respectively). PEMS-BAY covers the San Francisco Bay Area and METR-LA focuses on the traffic data of the Los Angeles Metropolitan Area. All these datasets collect three features (flow, occupation, and speed) at each location point over a period of time (with 5-minute time intervals). The spatial connection network for each dataset is constructed using the existing road network. The details of data statistics are shown in Table. 2. To use these datasets in experiments, we pre-process the features by z-score normalization.

Table 2: Basic statistics of the datasets used in our experiments.

| Data | PEMS03 | PEMS04 | PEMS07 | PEMS08 | PEMS-BAY | METR-LA |
|---|---|---|---|---|---|---|
| Location | CA, USA | | | | | |
| Time Span | 9/1/2018 - 11/30/2018 | 1/1/2018 - 2/28/2018 | 5/1/2017 - 8/31/2017 | 7/1/2016 - 8/31/2016 | 1/1/2017 - 5/31/2017 | 3/1/2012 - 6/30/2012 |
| Time Interval | 5 min | | | | | |
| Sensors | 358 | 307 | 883 | 170 | 325 | 207 |
| Edges | 547 | 340 | 866 | 295 | 2,369 | 1,515 |
| Time Steps | 26,208 | 16,992 | 28,224 | 17,856 | 52,116 | 34,272 |

### 5.2 Evaluation Metrics

We use Mean Absolute Error (MAE), Mean Absolute Percentage Error (MAPE), and Root Mean Squared Error (RMSE) metrics to evaluate the spatio-temporal forecasting. These metrics are defined as follows.

$$MAE = \frac{1}{n}\sum_{i=1}^{n}|y_i - \hat{y}_i|, \quad MAPE = (\frac{1}{n}\sum_{i=1}^{n}|\frac{y_i - \hat{y}_i}{y_i}|) * 100\%, \quad RMSE = \sqrt{\frac{1}{n}\sum_{i=1}^{n}\left(y_i - \hat{y}_i\right)^2},$$

### 5.3 Baselines

We compared our proposed GRAM-ODE with the following baselines.

- **ARIMA** (Box & Pierce, 1970): Auto-Regressive Integrated Moving Average is one of the most well-known statistical models for time-series analysis.

- **DCRNN** (Veeriah et al., 2015): Diffusion Convolutional Recurrent Neural Network utilizes diffusion graph convolutional networks with bidirectional random walks on directed graphs, and seq2seq gated recurrent unit (GRU) to capture spatial and temporal dependencies, respectively.

- **STGCN** (Yan et al., 2018): Spatio-Temporal Graph Convolutional Network combines graph structure convolutions with 1D temporal convolutional kernels to capture spatial dependencies and temporal correlations, respectively.

- **GraphWaveNet** (Wu et al., 2021): GraphWaveNet integrates adaptive graph convolution with 1D dilated casual convolution to capture spatio-temporal dependencies.

- **STSGCN** (Song et al., 2020): Spatio-Temporal Synchronous Graph Convolutional Networks decompose the problem into multiple localized spatio-temporal subgraphs, assisting the network in better capturing of spatio-temporal local correlations and consideration of various heterogeneities in spatio-temporal data.

---

[1]Datasets are downloaded from STSGCN github repository `https://github.com/Davidham3/STSGCN/`

Table 3: Performance comparison of GRAM-ODE and baselines on six benchmark datasets. A lower MAE/MAPE/RMSE indicates better performance. The **best** results are in bold and the second-best are underlined.

| Dataset | Metric | ARIMA | DCRNN | STGCN | GraphWaveNet | STSGCN | STFGNN | STGODE | **GRAM-ODE** |
|---|---|---|---|---|---|---|---|---|---|
| PEMS03 | MAE | 33.51 | 18.18 | 17.48 | 19.85 | 17.48 | 16.77 | 16.50 | **15.72** |
| | MAPE(%) | 33.78 | 18.91 | 17.15 | 19.31 | 16.78 | 16.30 | 16.69 | **15.98** |
| | RMSE | 47.59 | 30.31 | 30.12 | 32.94 | 29.21 | 28.34 | 27.84 | **26.40** |
| PEMS04 | MAE | 33.73 | 24.70 | 22.70 | 25.45 | 21.19 | 20.84 | 20.84 | **19.55** |
| | MAPE(%) | 24.18 | 17.12 | 14.59 | 17.29 | 13.90 | 13.02 | 13.77 | **12.66** |
| | RMSE | 48.80 | 38.12 | 35.55 | 39.70 | 33.65 | 32.51 | 32.82 | **31.05** |
| PEMS07 | MAE | 38.17 | 25.30 | 25.38 | 26.85 | 24.26 | 23.46 | 22.99 | **21.75** |
| | MAPE(%) | 19.46 | 11.16 | 11.08 | 12.12 | 10.21 | **9.21** | 10.14 | 9.74 |
| | RMSE | 59.27 | 38.58 | 38.78 | 42.78 | 39.03 | 36.60 | 37.54 | **34.42** |
| PEMS08 | MAE | 31.09 | 17.86 | 18.02 | 19.13 | 17.13 | 16.94 | 16.81 | **16.05** |
| | MAPE(%) | 22.73 | 11.45 | 11.40 | 12.68 | 10.96 | 10.60 | 10.62 | **10.58** |
| | RMSE | 44.32 | 27.83 | 27.83 | 31.05 | 26.80 | 26.22 | 25.97 | **25.17** |
| PEMS-BAY | MAE | 3.38 | 2.07 | 2.49 | 1.95 | 2.11 | 2.02 | 2.30 | **1.67** |
| | MAPE(%) | 8.30 | 4.90 | 5.79 | 4.61 | 4.96 | 4.79 | 4.61 | **3.83** |
| | RMSE | 6.50 | 4.74 | 5.69 | 4.48 | 4.85 | 4.63 | 4.89 | **3.34** |
| METR-LA | MAE | 6.90 | 3.60 | 4.59 | 3.53 | 3.65 | 3.55 | 3.75 | **3.44** |
| | MAPE(%) | 17.40 | 10.50 | 12.70 | 10.01 | 10.67 | 10.56 | 10.26 | **9.38** |
| | RMSE | 13.23 | 7.60 | 9.40 | 7.37 | 7.81 | 7.47 | 7.37 | **6.64** |

- **STFGNN** (Li & Zhu, 2021): Spatio-Temporal Fusion Graph Neural Networks uses Dynamic Time Warping (DTW) algorithm to gain features, and follow STSGCN (Song et al., 2020) in using sliding window to capture spatial, temporal, and spatio-temporal dependencies.

- **STGODE** (Fang et al., 2021): Spatio-Temporal Graph ODE Networks attempt to bridge continuous differential equations to the node representations of road networks in the area of traffic forecasting.

## 5.4 Experimental Settings

Following the previous works in this domain, we perform experiments by splitting the entire dataset into 6:2:2 for train, validation, and test sets. This split follows a temporal order, using the first 60% of the time length for training, and the subsequent 20% each for validation and testing. We use the past one hour to predict the future one hour. Since the time interval of data collection is 5 minutes, $L, L' = 12$ temporal data points. Based on Table 2, we have different number of sensors among different datasets, therefore, $|V|$ is different. DTW threshold ($\epsilon$) in Eq. (2) is 0.1; number of channels ($C$) in the historical data is 3 (i.e., flow, speed, and occupation) and in the embedding space is 64. The shared temporal weights $W_{s1}, W_{s2} \in \mathbb{R}^{12 \times 12}$ are initialized randomly from normal distribution. The length of latent space for the input of local ODE block is $L'' = 4$, and in the final attention module, number of attention heads $h = 12$. During training, we use the learning rate of $10^{-4}, 10^{-4}, 10^{-5}, 10^{-5}, 10^{-4}$, and $10^{-4}$ for PEMS03, PEMS04, PEMS07, PEMS08, PEMS-BAY, and METR-LA datasets, respectively. The optimizer is AdamW. All experiments are implemented using PyTorch (Paszke et al., 2019) and trained using Quadro RTX 8000 GPU, with 48GB of RAM.

## 5.5 Experimental Results

Our proposed GRAM-ODE outperforms other baselines on all datasets in Table 3, except for the MAPE metric in PEMS07, where it is slightly greater than that of STFGNN. ARIMA performs considerably worse than other baselines, likely because it ignores the graph structure of spatio-temporal data. GraphWaveNet performs relatively poorly, possibly due to its limited capability in stacking spatio-temporal layers and expanding the receptive field learned from 1D CNN temporal kernels. DCRNN uses bi-directional random walks and a GRU to model spatial and temporal information, respectively, but its relatively low modeling efficacy and efficiency for long-range temporal information may contribute to its subpar performance. STGCN uses GCN for the spatial domain and 1D dilated convolutional operations for the temporal domain, but may lose local information due to the dilation operation, while its absence of attention-based operations and limited capability of convolutional operations in modeling high-dimensional spatial, temporal, and spatio-temporal correlations may also result in relatively poor performance. STSGCN captures local correlations with localized spatio-temporal subgraphs but may miss global information and thus perform poorly in long-range forecasting and with data containing missing entries or noise. STFGNN uses DTW for latent spatial networks and performs well, but is limited in learning comprehensive spatio-temporal correlations. STGODE

uses Neural ODEs to capture spatio-temporal dynamics and achieves very good performance compared to other baselines, but still lacks the ability to capture complex spatio-temporal correlations, balance local and global patterns, and model dynamic interactions between its components.

## 5.6 Case Study

We selected two nodes from the PEMS08 road networks to conduct our case study for the qualitative demonstration of results. As Fig.4 shows, the predicted curves by our proposed GRAM-ODE (red curve) is better (more closely) aligned with the ground truth than STGODE (grey curve). The ground truth in node 109 has more fluctuations compared to the node 17 which causes more difficulty in the forecasting task. We can also observe that our model provides faster response in case of these abrupt fluctuations. This highlights the effectiveness of local ODE-GNN block in our model architecture which helps the model to better learn the local fluctuations.

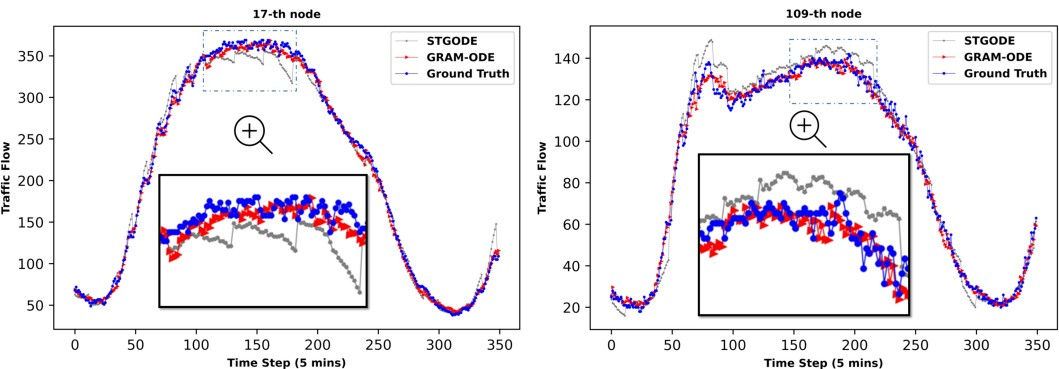

Figure 4: The comparison of traffic flow forecasting between our proposed GRAM-ODE and STGODE visualized for node 17 (left column) and node 109 (right column) of the PEMS08 dataset.

## 5.7 Ablation Study

To investigate the effect of different components of GRAM-ODE, we conduct ablation experiments on PEMS04 and PEMS08 with several different variants of our model.

(1) **Base:** In this model, data only passes through a TCN, an ODE block and another TCN module. The ODE block only contains the global ODE-GNN with a fully connected layer after that. The output of TCN is then imported to a simple fully connected layer instead of an attention module.

(2) **+E:** Beyond (1), this model adds the dynamic edge correlations with an edge-based ODE-GNN block and aggregates its outputs with the global ODE-GNN block through a simple weighted sum operation.

(3) **+L:** Beyond (2), this model adds the local ODE-GNN block with different temporal kernels to better consider the local temporal patterns. The outputs of local ODE modules are aggregated with other ODE modules through a weighted sum operation.

(4) **+share:** Compared to (3), this model uses shared temporal weights between the edge and global ODE-GNN modules and uses shared spatial weights among all three ODE-GNN modules, global, local and edge module.

(5) **+cons:** Beyond (4), this model adds an adaptive message filtering constraint to restrict the divergence of embeddings from local and global ODE modules.

(6) **+agg:** Beyond (5), this model replaces the weighted sum aggregation with a newly designed aggregation module explained in Eq. (19).

(7) **+res:** Beyond (6), this model adds the intra-block residual connections between outputs and inputs of the multi ODE-GNN blocks (which is explained in Eq. (20)).

(8) GRAM-ODE: Beyond (7), this model replaces the last linear layer with the attention module to better combine the features learned from different traffic graphs and parallel channels of GRAM-ODE layers (given by Eqs. (22) and (23)).

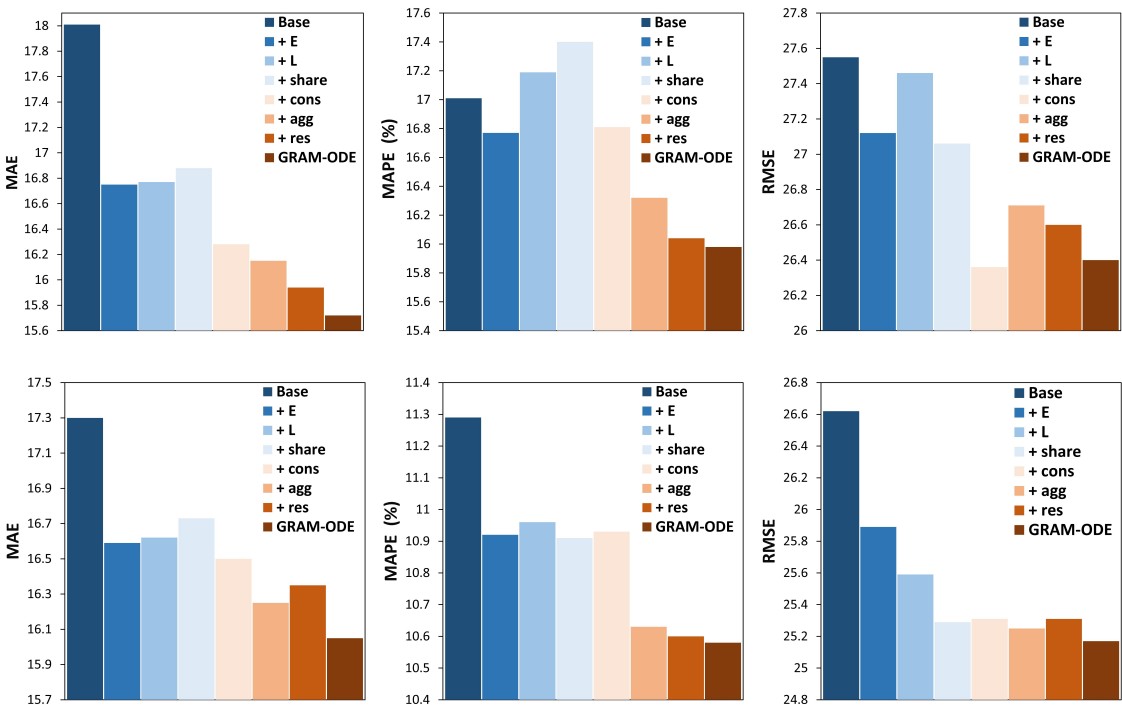

Figure 5: Ablation experiment results with different configurations of GRAM-ODE on PEMS03 (top row) and PEMS08 (bottom row) datasets.

Fig. 5 shows the results of ablation experiments with MAE, MAPE, and RMSE metrics. It can be observed that the edge and local ODE-GNN modules are both enhancing the feature representation in the traffic forecasting task. The model variant '+E' improves the performance of the base model across all metrics and both datasets. This shows that the simple node-based global ODE-GNN is insufficient in learning informative features and adding the edge-based global ODE-GNN module can considerably help the performance. Without any further techniques, the model gets worse by only using '+L' beyond '+E'. However, after adding the shared weight and divergence constraint techniques, the model usually gets better across all metrics. The shared weights are applied in the spatial and temporal operations of global node-based and edge-based ODE-GNN blocks to take into account the dynamic correlations of edges as well as nodes in the model formulation. The constraint is added to prevent local and global ODE-GNN embeddings deviating from one another. In this figure, we can also observe the impact of aggregation layer, residual-based update layer as well as the designed attention module. It appears that, among these three elements, adding the attention module ($AM$) will always result in better performance, which is consistent with our hypothesis that the attention mechanism makes the model more effective in considering all the high-dimensional correlations during feature extraction.

## 5.8 Robustness Study

To evaluate the robustness of our model, we add noise to the historical input of training data, which can potentially mimic uncertainties and biases that can arise during the data collection process. The added noise follows zero-mean i.i.d. Gaussian distribution with fixed variance, e.g., $\mathcal{N}(0, \gamma^2)$, where $\gamma^2 \in \{2, 4\}$. We conduct robustness analysis across different values of $n \in \{0.1, 0.2, 0.3, 0.4\}$ representing the ratio of training data impacted by noise. In other words, $n = 0$ captures the performance without any noise. Fig. 6 represents the results of robustness comparisons between GRAM-ODE and STGODE on PEMS04 dataset across all the three metrics (MAE, MAPE, and RMSE). It can be observed that GRAM-ODE performance is more robust than STGODE with different levels of added noise $n = 0.1; 0.2; 0.3; 0.4$ which is probably due to the powerful spatio-temporal feature extraction gained by multiple ODE-GNN modules. We can also notice that, when the noise levels are high (with $\gamma^2 = 4$ and $n = 0.4$), GRAM-ODE can still beat many baseline models listed in Table 3, demonstrating the significant benefit of incorporating various ODE modules in our framework which can improve robustness.

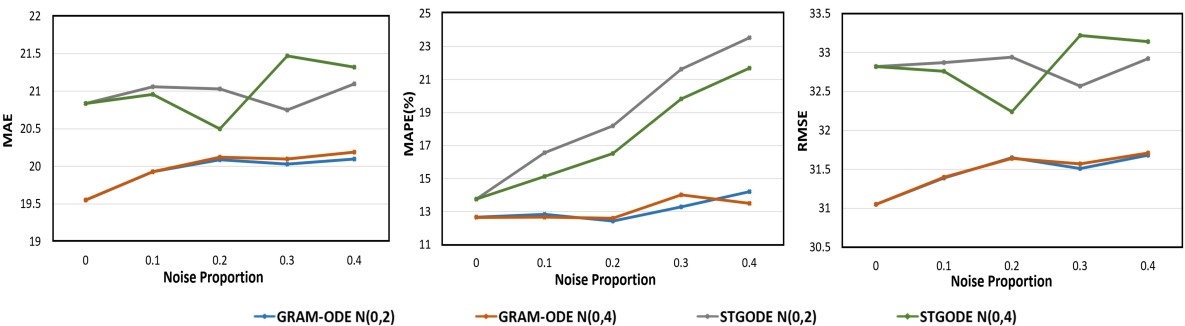

Figure 6: Robustness comparison of GRAM-ODE and STGODE on PEMS04 dataset.

## 6  Conclusion

In this paper, we propose Spatio-Temporal Graph Multi-ODE Neural Networks (GRAM-ODE) for forecasting traffic. In this model, multiple coupled ODE-GNN blocks are used to capture complex spatio-temporal dependencies from different views and learn better representations. We also add some techniques to further improve the communication between different ODE-GNN blocks including sharing weights, advanced aggregation, and divergence constraint. Extensive experiments on six real-world datasets show the superior performance of our proposed model compared to other state-of-the-art models as well as the effectiveness of each component. Future work may focus on investigating model compression techniques to reduce model size without sacrificing performance, exploring distributed computing strategies for efficiency, and evaluating GRAM-ODE's applicability to other spatio-temporal applications like climate modeling or social network analysis.

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

# Appendix

## A   Prediction Variance

### A.1   Initialization Randomness

Table 4 presents the average and standard deviation results of multiple runs of our model using five different random seeds. Results demonstrate that GRAM-ODE's performance exhibits only minor variance across different seeds. Moreover, despite these variations, GRAM-ODE continues to outperform other baselines across different benchmark datasets, as also evident by comparing to baseline results in Table 3.

Table 4:   GRAM-ODE performance variance for different random seeds over all datasets.

| Dataset | Metric | GRAM-ODE |
|---|---|---|
| PEMS03 | MAE | $15.72 \pm 0.29$ |
| | MAPE(%) | $15.98 \pm 0.31$ |
| | RMSE | $26.40 \pm 0.42$ |
| PEMS04 | MAE | $19.55 \pm 0.12$ |
| | MAPE(%) | $12.66 \pm 0.29$ |
| | RMSE | $31.05 \pm 0.25$ |
| PEMS07 | MAE | $21.75 \pm 0.30$ |
| | MAPE(%) | $9.74 \pm 0.21$ |
| | RMSE | $34.42 \pm 0.39$ |
| PEMS08 | MAE | $16.05 \pm 0.20$ |
| | MAPE(%) | $10.58 \pm 0.13$ |
| | RMSE | $25.17 \pm 0.14$ |
| PEMS-BAY | MAE | $1.67 \pm 0.02$ |
| | MAPE(%) | $3.83 \pm 0.03$ |
| | RMSE | $3.34 \pm 0.03$ |
| METR-LA | MAE | $3.44 \pm 0.08$ |
| | MAPE(%) | $9.38 \pm 0.11$ |
| | RMSE | $6.64 \pm 0.05$ |

### A.2   Training Randomness

We also conducted experiments using three distinct random seeds across various cross-validation splits on the PEMS-BAY dataset (due to time constraints, we were only able to run this experiment for one dataset). The results of these experiments can be found in Table 5. As stated in the manuscript, we follow the previous works and divide the datasets into train/val/test splits with a 6:2:2 ratio. Given that the data is time-series, we can create different cross-validation splits for the train and test data as follows: T_X, TX_, _TX, XT_, _XT, X_T where T and X refers to train and test sets. Despite the minor variance in performance across different cross-validation splits and model initialization seeds, we can observe that GRAM-ODE still outperforms other baselines for this dataset.

## B   Efficiency and Scalability

Figure 7 comprises two subfigures that demonstrate the efficiency and scalability of our proposed GRAM-ODE. Fig. 7(a) presents a Pareto plot illustrating the trade-off between RMSE forecasting performance and inference time for various methods, including our GRAM-ODE. Although our multi-ODE framework exhibits a larger inference time, it delivers superior RMSE performance compared to other methods. This highlights that, in use cases where prediction accuracy is of paramount importance, the enhanced performance of our

Table 5: GRAM-ODE performance variance for different cross-validation splits across different model initialization seeds on PEMS-BAY dataset.

| CV Split | RMSE | MAE | MAPE | Seed ID |
|---|---|---|---|---|
| T,__,X | 3.39 | 1.7 | 3.95 | 0 |
|  | 3.39 | 1.69 | 3.91 | 1 |
|  | 3.39 | 1.7 | 3.92 | 2 |
| T,X,__ | 3.38 | 1.69 | 3.97 | 0 |
|  | 3.39 | 1.68 | 3.96 | 1 |
|  | 3.38 | 1.68 | 3.94 | 2 |
| __,T,X | 3.26 | 1.62 | 3.67 | 0 |
|  | 3.24 | 1.6 | 3.63 | 1 |
|  | 3.27 | 1.62 | 3.68 | 2 |
| X,T,__ | 2.97 | 1.59 | 3.44 | 0 |
|  | 3.00 | 1.63 | 3.49 | 1 |
|  | 2.98 | 1.59 | 3.45 | 2 |
| __,X,T | 3.41 | 1.82 | 4.05 | 0 |
|  | 3.4 | 1.8 | 4.01 | 1 |
|  | 3.41 | 1.82 | 4.03 | 2 |
| X,__,T | 3.18 | 1.72 | 3.79 | 0 |
|  | 3.21 | 1.75 | 3.83 | 1 |
|  | 3.18 | 1.71 | 3.78 | 2 |

GRAM-ODE can outweigh the increased computational complexity. Fig. 7(b) showcases the percentage performance gain achieved by GRAM-ODE in comparison to the best-performing baseline. The dataset size on the x-axis is calculated by multiplying the squared number of nodes by the number of timesteps. As the dataset size increases, we observe that GRAM-ODE provides relatively better performance gains, indicating that our multi-ODE framework is scalable to larger datasets and higher-dimensional inputs. This scalability further emphasizes the advantages of GRAM-ODE, especially when handling more extensive and complex data.

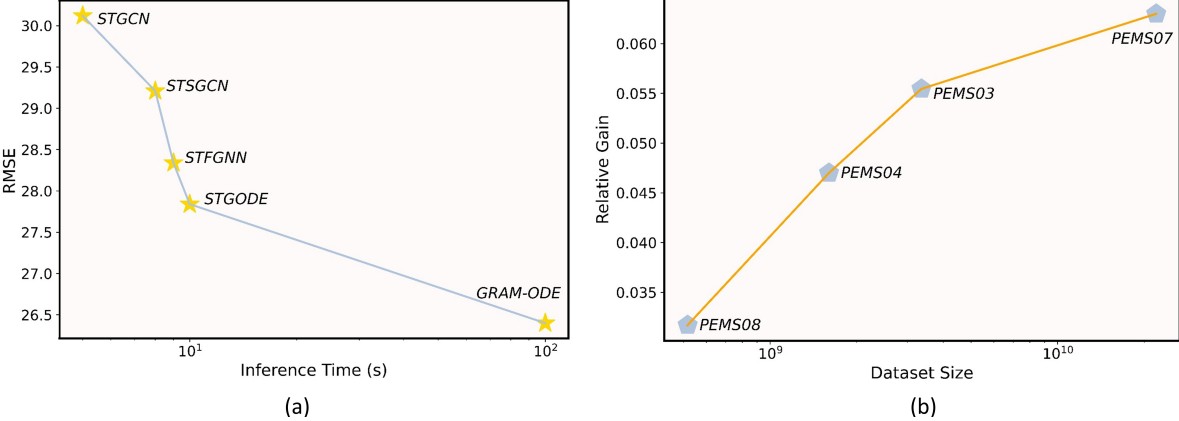

(a)  (b)

Figure 7: Efficiency and Scalability of GRAM-ODE. (a) RMSE and inference time trade-off for various methods on the PEMS03 dataset. (b) Relative RMSE gain by GRAM-ODE in comparison to the best-performing baseline across datasets of varying sizes.

## C  Exploring the Outputs of Various ODE Modules

We analyzed the outputs of these different ODE modules in the aftermath of a traffic jam event, when fluctuations and changes are more pronounced. Our observations suggest that the distinct ODE modules

contribute to varying performance and predictions, likely resulting from the different features they learn from multiple perspectives. In particular, Fig. 8 shows that shortly after the traffic jam, when traffic flow have sudden increases, the outputs of the Local Module (LM) appear to align more closely with the ground truth. As traffic flow stabilizes, both the Local Module (LM) and Global Module (GM) outputs exhibit better alignment with the ground truth. However, as traffic flow gradually decreases, the Global Module (GM) outputs start to diverge from the ground truth, while the Edge Module (EM) and Local Module (LM) outputs remain more consistent with the actual data. Therefore, as traffic flow changes, the alignment of the outputs with the ground truth varies across modules, indicating that each module captures different aspects of the data.

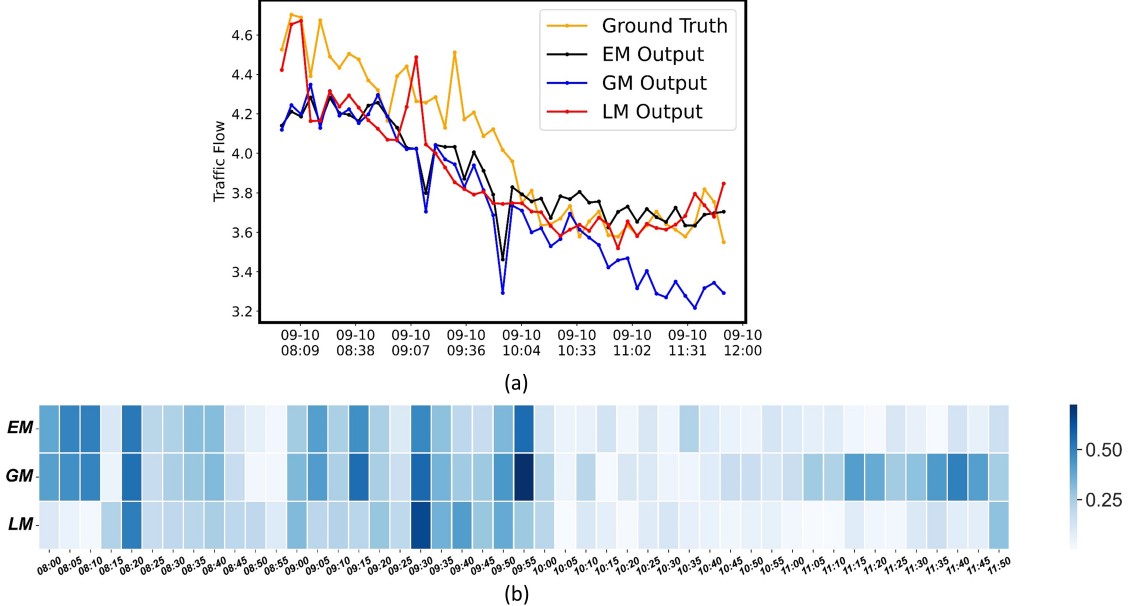

Figure 8: Outputs of different ODE modules during traffic fluctuations. (a) Varying performance and predictions of the Local Module (LM), Glocal Module (GM), and Edge Module (EM) in response to changes in traffic flow. (b) Normalized distance of different prediction curves with ground truth at each time step.

