# OpenReview forum: "Graph-based Multi-ODE Neural Networks for Spatio-Temporal Traffic Forecasting"
_TMLR — Accepted by TMLR_

### Review · Reviewer_uwEo · 2023-01-09

**Summary Of Contributions:**

This paper introduces Graph-based Multi-ODE (GRAM-ODE), a neural network architecture for spatio-temporal traffic forecasting. The model consists of novel components for addressing temporal patterns at different scales (e.g. short-term vs. long-term) and for including semantic edge information. The model further uses multi-head attention pooling to aggregate features (while prior work in this area primarily used simpler aggregation modules such as sum- or max-pooling). The method is validated on four traffic prediction datasets and shows benefits over prior works.

**Audience:**

Yes

**Broader Impact Concerns:**

Not applicable. The work is an incremental architectural improvement on an existing task and model, i.e. no Broader Impact Statement is required.

**Claims And Evidence:**

Yes

**Requested Changes:**

* [Critical] The experimental protocol needs to be revisited: the authors should not only analyze variance of results across the different cross-validation splits, but should also report variance for different model initialization seeds, i.e. how much variance the results have based on randomness in model training/initialization. Only results that are significantly different from baselines or model variants (in terms of ablations) can be considered for testing the initial hypotheses made in the paper.
* [Critical] The claims and hypotheses that the paper proposes need to be revisited after improving the experimental protocol. It is quite likely that several claims made in the current version of the paper will not survive this test. For example, already in the current version of the paper, the effect of special handling for local temporal patterns (“+L” ablation) is minimal or even negative.
* [Critical] The clarity of the model definition and figures need to be significantly improved. I suggest a major revision of the method section, in line with the potential revision of the claims/hypotheses.
* [Highly recommended] The authors should consider a wider range of datasets and/or tasks to more convincingly test their hypotheses and to show more evidence for their claims.
* [Highly recommended] The authors should consider reducing the length of the paper to 12 pages (excl. appendix) or less; the current version of the paper is far less concise than it could be. Improving conciseness would also significantly help with clarity issues that the paper currently has.


Other question:
Regarding the shared spatial weight: as this is a dense matrix (with parameters initialized from a Gaussian distribution) that is being added to the adjacency matrix, this means that all message passing operations will be dense thereafter. This seems to be infeasible for large graphs, which would generally be the case for real-world applications of traffic modeling. Could the authors comment on this?


**Strengths And Weaknesses:**

**Strengths:**
* Novel architectural contributions are reasonably well-motivated
* Apparent (but potentially marginal) improvements on an existing traffic prediction task
* Detailed ablation study and study of model robustness to noise

**Weaknesses:**
* Experimental protocol does not support claims/hypotheses well: no variance is reported; datasets are of very limited diversity/size
* Insufficient clarity and quality of writing and exposition

**Regarding clarity of writing:**

Figures:
* Figure 2 is unclear. It is difficult to get an overview of the model.
* Figure 3 is similarly unclear. What do the blue rectangles represent? What does “AM” stand for? What does it mean if a rectangle is shaded in a particular way (some are colored with a gradient from dark blue to blue, whereas others are from light blue to white)?

Writing:
* “To fuse these two separate sets of operations more intelligently” -> this is a vague statement: how would one determine whether a particular fusion operation is more or less intelligent?
* The paper goes straight into very specific details around the newly introduced weight sharing, before introducing an overview of the model: I would recommend that the paper starts with a concise overview of the model and then introduces the particular novel components.

Lastly, there are frequent typos and grammatical inconsistencies in the paper. A few examples below:
* Table 1 (notations): inconsistent lower-/upper-casing; “numbers of channels”
* “Identical matrix” -> “identity matrix”


**Regarding experiments:**

Experiments:
* Results look largely comparable to prior work. It is unclear whether benefit stems from a better choice of hyperparameters or from modeling novelties
* No error bars or indication of variance
* 4 small datasets from the same provider, i.e. it is unclear how this model behaves on vastly different road networks, e.g. of larger size, in different continents, cities etc.

---

> ### Author Response · Authors · 2023-03-29
> **Response to Reviewer-uwEo**
>
> We would like to thank the reviewer uwEo for the insightful comments. Below, we address your concerns and provide clarification.
>
> ---
> > Comment:
> > * Figure 2 is unclear. It is difficult to get an overview of the model.
> > * Figure 3 is similarly unclear. What do the blue rectangles represent? What does “AM” stand for? What does it mean if a rectangle is shaded in a particular way (some are colored with a gradient from dark blue to blue, whereas others are from light blue to white)?
> > * To fuse these two separate sets of operations more intelligently” -> this is a vague statement: how would one determine whether a particular fusion operation is more or less intelligent?
> > * The paper goes straight into very specific details around the newly introduced weight sharing, before introducing an overview of the model: I would recommend that the paper starts with a concise overview of the model and then introduces the particular novel components.
> > * Table 1 (notations): inconsistent lower-/upper-casing; “numbers of channels
> > * Identical matrix” -> “identity matrix
> > * The clarity of the model definition and figures need to be significantly improved. I suggest a major revision of the method section, in line with the potential revision of the claims/hypotheses.
>
> **Response**:
> We appreciate the reviewer for the constructive feedback. We carefully consider your suggestions and update the manuscript accordingly. Here are some of the updates:
> * We have revised both Figure 2 and Figure 3 to improve clarity. Regarding Figure 3, the yellow, orange and green rectangles represent outputs of different ODE modules (edge, local and global, respectively). We have now included a more detailed captions for this figure to clarify their meaning. The abbreviation "AM" in both figures refers to the Attention Module, as defined in Section 4.2. We have also clarified this point in the revised manuscript.
> * We would like to thank you for pointing out the vague statements and inconsistencies. We have rephrased the vague statements and addressed inconsistencies in the revision.
> * We have restructured the methodology section to start with an overview of the GRAM-ODE model (Section 4 Overview and Figure 2), followed by a detailed discussion of its individual components.
>
> ---
> > Comment:
> > * The authors should consider a wider range of datasets and/or tasks to more convincingly test their hypotheses and to show more evidence for their claims.
> > * The experimental protocol needs to be revisited: the authors should not only analyze variance of results across the different cross-validation splits, but should also report variance for different model initialization seeds, i.e. how much variance the results have based on randomness in model training/initialization. Only results that are significantly different from baselines or model variants (in terms of ablations) can be considered for testing the initial hypotheses made in the paper.
>
> **Response**:
> We thank the reviewer for bringing up this. In response to your comments, we have conducted additional experiments and updated the manuscript accordingly:
> In order to demonstrate the wider applicability of the proposed model and the generalizability of the performance improvements across different kinds of data, we added two new benchmark datasets for traffic forecasting: **PEMS-BAY** and **METR-LA**. The data statistics for these datasets are added to Table 2 in the manuscript. Notably, these two datasets differ from the previously evaluated datasets in terms of edge sparsity, as they are less sparse and have a greater number of edges. All the six evaluaton datasets that we used in this study are standard and widely used benchmark datasets for the traffic forecasting task. We will update Table 3 in the manuscript with the results of experiments on the new datasets, which show that our method, GRAM-ODE, also performs better on these additional datasets.
> We also performed multiple runs of our model for different random seeds, and reporting the average and standard deviation results. The standard deviation for other baselines is not reported, as their results is adopted from their original papers, which did not include this information. Please refer to Appendix (section B.1, Table 4), which now includes results of GRAM-ODE with standard deviations over all six datasets.
>
> We have conducted experiments using three distinct random seeds across various cross-validation splits on the PEMS-BAY dataset (due to time constraints, we were only able to run this experiment for one dataset). The results of these experiments can be found in Appendix (section B.2, Table 5). As stated in the manuscript, we follow the previous works and divide the datasets into train/val/test splits with a 6:2:2 ratio. Given that the data is time-series, we can create different cross-validation splits for the train and test data as follows: T\_X, TX\_, \_TX, XT\_, \_XT, X\_T where T and X refer to train and test sets.

---

> > ### Author Response · Authors · 2023-03-29
> > **Response to Reviewer-uwEo (2)**
> >
> > ---
> > > Comment:
> > > *  The claims and hypotheses that the paper proposes need to be revisited after improving the experimental protocol. It is quite likely that several claims made in the current version of the paper will not survive this test. For example, already in the current version of the paper, the effect of special handling for local temporal patterns (“+L” ablation) is minimal or even negative.
> >
> > **Response**:
> > We agree with the importance of considering potential performance variance in GRAM-ODE due to randomness in model training and initialization. In line with the new experimental protocol suggested by the reviewer, we have observed that GRAM-ODE's performance indeed exhibits minor variance across different cross-validation splits and model initialization seeds. Nevertheless, even accounting for these variations, GRAM-ODE continues to outperform other baselines across different benchmark datasets.
> >
> > Regarding the minimal or negative effect of the local temporal pattern in the ablation study, we would like to clarify that the model's ablation variants (detailed in Section 5.7) are added incrementally. The “+L” variant demonstrates the performance of the model solely after incorporating a local ODE-GNN block without any connection with other ODE-GNN blocks, emphasizing the importance and necessity of such connections between various ODE-GNN blocks in the model. Our results indicate that the inclusion of connection techniques, such as shared weights (temporal and spatial) and message filtering constraints, between different ODE-GNN blocks significantly enhances the model's performance.
> >
> > ---
> > > Comment:
> > > *  Other question: Regarding the shared spatial weight: as this is a dense matrix (with parameters initialized from a Gaussian distribution) that is being added to the adjacency matrix, this means that all message passing operations will be dense thereafter. This seems to be infeasible for large graphs, which would generally be the case for real-world applications of traffic modeling. Could the authors comment on this?
> >
> > **Response**:
> > We thank the reviewer for bringing up this important point. It is true that adding the shared spatial dense matrix to the adjacency matrix makes message passing operations dense, and computation may grow exponentially for large graphs. To handle this, the model uses localized convolution operations which limit each node to only receive messages from its local neighborhood, thus, reducing the number of parameters in the message passing process.
> >
> > ---
> > > Comment:
> > > *  The authors should consider reducing the length of the paper to 12 pages (excl. appendix) or less; the current version of the paper is far less concise than it could be. Improving conciseness would also significantly help with clarity issues that the paper currently has.
> >
> > **Response**:
> > Thank you for the suggestion. We have revised the paper with your suggestions in mind and ensure that the revised version is as clear and concise as possible.

---

> > > ### Comment · Reviewer_uwEo · 2023-04-07
> > > **Re: Response to Reviewer-uwEo**
> > >
> > > Thank you for addressing my comments in great detail. My concerns around the experimental evaluation have been addressed, I appreciate testing the method on additional datasets and reporting the variance of results across different model initializations.
> > >
> > > I find the clarity of the paper in the revised draft to be improved as well, although my concerns around the clarity of the figures still remains to some degree: the figures still contain a lot of details that are difficult to understand (the improved captions certainly help!), and I still find them somewhat unhelpful to understand the method. This is not critical for someone with sufficient time to understand the details of the method, since the model is explained in the main text, but it certainly still makes the paper somewhat difficult to read. I also think it would be valuable for the authors to condense the paper into 12 pages and to put a more concise focus on the core aspect. The current length of the paper is not necessary, in my view.
> > >
> > > Overall, this is a borderline paper that can be accepted. I think the paper could have a higher impact if the authors would spend more effort on improving clarity and presentation, while keeping the paper at 12 pages or less.

---

> > > > ### Author Response · Authors · 2023-04-10
> > > > **Response to Reviewer-uwEo (3)**
> > > >
> > > > We appreciate your feedback and are pleased to know that our revisions have addressed your concerns regarding experimental evaluation.
> > > > We will refine and simplify the figures in the final version of paper to improve their clarity further. Regarding the paper's length, we have condensed it by approximately two pages in the last submission. While we recognize your rationale for a 12-page limit, we believe the additional information, particularly the methodological details such as notation table, figures, and algorithms, serves to facilitate a deeper understanding of the method for readers. Considering that TMLR does not impose a strict page limit, we prefer to provide comprehensive information to ensure clarity.
> > > > We will be happy to make changes if deemed necessary.

---

### Review · Reviewer_LuiQ · 2023-01-24

**Summary Of Contributions:**

This paper focuses on the task of spatio-temporal traffic forecasting. Given the recent advances of graph ODE and their limitations on the task, the authors proposed multi-ODE network (GRAM-ODE). The proposed method seems to be able to capture complex local and global dynamics from the spatial-temporal dependencies.

**Audience:**

Yes

**Claims And Evidence:**

Yes

**Requested Changes:**

please address the concerns in the previous Q.

**Strengths And Weaknesses:**

This paper focuses on the task of spatio-temporal traffic forecasting. Given the recent advances of graph ODE and their limitations on the task, the authors proposed multi-ODE network (GRAM-ODE). The proposed method seems to be able to capture complex local and global dynamics from the spatial-temporal dependencies. Following are my concerns about this work, which I hope the authors can address (some of which are minor concerns):

1. I would recommend the authors to add some citations in the first sentence of the Introduction.

2. The different arrows in fig. 1 is barely readable.

3. For the results showed in Table 3, I would suggest the authors to also report the standard deviations for a better understanding on the significances. Ideally, averaging the results of multiple runs of cross validation and reporting the average with standard deviation would be the best.

4. Given the audience of this journal, I think Eq. 25 might not be necessary.

5. I wonder if there are other publicly available datasets that can be used for evaluation, which can add more diversity to the datasets.

6. Given the multi-ODE design of the proposed method, I think it's necessary to also compare with baselines on perspectives of scalability and efficiency. For example, complexity analysis of the proposed method, runtime comparison of the proposed method against baselines, etc.

7. In the introduction, the authors mentioned that different ODE modules proposed in this work were designed for capturing different knowledge such as local temporal patterns or semantic edges. How do we validate these claims? While there are experiments showing better overall performances and ablation studies showing each designed components contributing to the performance, I think it is still hard to conclude that one design is specifically capturing some knowledge.

8. Some of the different colors in Fig. 5 are barely separable, please consider changing to more separable colors or patterns for better readability.

9. The manuscript seems is missing a careful proofread and typo check.

---

> ### Author Response · Authors · 2023-03-29
> **Response to Reviewer-LuiQ**
>
> We would like to thank the reviewer LuiQ for the insightful comments. Below, we address your concerns and provide clarification on specific points.
>
>
> ---
> > Comment:
> > * I would recommend the authors to add some citations in the first sentence of the Introduction.
> > * The different arrows in fig. 1 is barely readable.
> > * Some of the different colors in Fig. 5 are barely separable, please consider changing to more separable colors or patterns for better readability.
> > * The manuscript seems is missing a careful proofread and typo check.
>
>
> **Response**:
> Thank you for the comments. We have carefully considered your suggestions and revised the manuscript accordingly.
>
>
>
> ---
> > Comment:
> > * I wonder if there are other publicly available datasets that can be used for evaluation, which can add more diversity to the datasets.
> > * For the results showed in Table 3, I would suggest the authors to also report the standard deviations for a better understanding on the significances. Ideally, averaging the results of multiple runs of cross validation and reporting the average with standard deviation would be the best.
>
> **Response**:
> We thank the reviewer for bringing up this. In response to your comments, we have conducted additional experiments and updated the manuscript accordingly:
>
> In order to demonstrate the wider applicability of the proposed model and the generalizability of the performance improvements across different kinds of data, we added two new benchmark datasets for traffic forecasting: **PEMS-BAY** and **METR-LA**. The data statistics for these datasets are added to Table 2 in the manuscript. Notably, these two datasets differ from the previously evaluated datasets in terms of edge sparsity, as they are less sparse and have a greater number of edges. All the six evaluaton datasets that we used in this study are standard and widely used benchmark datasets for the traffic forecasting task. We have updated Table 3 in the manuscript with the results of experiments on the new datasets, which show that our method, GRAM-ODE, also performs better on these additional datasets.
>
> We performed multiple runs of our model for different random seeds, and reporting the average and standard deviation results in Appendix (section B.1, Table 4). The standard deviation for other baselines is not reported, as their results is adopted from their original papers, which did not include this information.
> We also conducted experiments using three distinct random seeds across various cross-validation splits on the PEMS-BAY dataset (due to time constraints, we were only able to run this experiment for one dataset). The results of these experiments can be found in Appendix (section B.2, Table 5). As stated in the manuscript, we follow the previous works and divide the datasets into train/val/test splits with a 6:2:2 ratio. Given that the data is time-series, we can create different cross-validation splits for the train and test data as follows: T\_X, TX\_, \_TX, XT\_, \_XT, X\_T where T and X refers to train and test sets.
>
>
> ---
> > Comment:
> > * In the introduction, the authors mentioned that different ODE modules proposed in this work were designed for capturing different knowledge such as local temporal patterns or semantic edges. How do we validate these claims? While there are experiments showing better overall performances and ablation studies showing each designed components contributing to the performance, I think it is still hard to conclude that one design is specifically capturing some knowledge.
>
> **Response**:
> We appreciate your concerns about validating our claim that different ODE modules capture distinct knowledge, such as local temporal patterns and dynamic semantic edges. While it is challenging to conclusively verify the specific knowledge captured by each design due to the inherent limitation of interpretability in neural networks, our ODE modules have been purposefully designed to achieve these goals. For instance, we utilize local temporal kernels and ODE modules for each local embedding to capture local temporal patterns. Similarly, we identify dynamic semantic edges based on node embeddings and model their patterns over time using a separate ODE module.
> To provide additional supporting evidence, we have analyzed the outputs of these different ODE modules after a traffic jam event, when fluctuations and changes are more pronounced (Appendix, section D, Figure 8). Our observations suggest that the distinct ODE modules contribute to varying performance and predictions, likely resulting from the different features they learn from multiple views. As traffic flow changes, the alignment of the outputs with the ground truth varies across modules, indicating that each module captures different aspects of the data.

---

> > ### Author Response · Authors · 2023-03-29
> > **Response to Reviewer-LuiQ (2)**
> >
> > ---
> > > Comment:
> > > * Given the multi-ODE design of the proposed method, I think it's necessary to also compare with baselines on perspectives of scalability and efficiency. For example, complexity analysis of the proposed method, runtime comparison of the proposed method against baselines, etc.
> >
> > **Response**:
> > Thank you for your insightful comment regarding the scalability and efficiency of our multi-ODE framework. We acknowledge that our method might be computationally more expensive than the baselines. However, we want to point out the superior performance of our multi-ODE framework, which can outweigh the increased complexity in certain use cases where prediction accuracy is of paramount importance. In response to your comment, we have taken the following steps:
> >
> >
> > * We present a performance vs. runtime pareto comparison between our proposed method and the baselines in the Appendix (section C, Figure 7(a)), highlighting the performance-efficiency trade-off for the traffic forecasting task.
> >
> > * We also analyze the scalability of our multi-ODE framework for larger datasets and higher-dimensional inputs in the Appendix (section C, Figure 7(b)). More discussion on the scalability of our method is included in the revised version of the paper
> >
> > * In the paper's future work section, we discuss possible strategies for enhancing our approach's efficiency, providing direction for further research to optimize the method for real-world applications.

---

> > > ### Comment · Reviewer_LuiQ · 2023-04-11
> > > **update**
> > >
> > > I appreciate the authors' detailed explanations as well as the additional experiments. I think my concerns have been addressed.

---

### Review · Reviewer_MPh2 · 2023-03-25

**Summary Of Contributions:**

This paper provides a graph convolutional network (GCN) for traffic forecasting, called GRAM-ODE. It claims the previous works ignore the local temporal embedding and edge-based dynamic embedding. Then it proposes the multi-ODE layer including the sublayers to process global temporal, local temporal and edge-based features, respectively. An attention-based aggregation layer is also applied to merge the advanced embeddings from two traffic graphs. The experiments show: 1) GRAM-ODE surpasses selective baselines; 2) the effectiveness of the key components in GRAM-ODE; 3) the robustness of the trained GRAM-ODE.

**Audience:**

Yes

**Broader Impact Concerns:**

I have no concerns on the ethical implications of the work.

**Claims And Evidence:**

Yes

**Requested Changes:**

The following points are necessary to be addressed for meeting the level of acceptance.

1. I strongly recommend the authors to present the analyses on the training and testing efficiency of the proposed approach.
2. The above highly related works should be added in the literature review or excluded by some reasons.
3. The experimental setting should be introduced with more words. The limitations and the future work should be added.



**Strengths And Weaknesses:**

**Strengths:** Traffic forecasting is a very interesting and challenging research topic. This paper has described the background of the topic, with a good motivation to make the graph embedding of the traffic network more informative. The proposed method is based on ordinary differential equation (ODE), which is a commonly used component in traffic forecasting. The technical part is fairly sound and not difficult to understand. The experiments contain various methods for traffic forecasting and the ablation study has verified the effect of the components. This work also considers the noisy data and evaluates the robustness of the proposed method, which I think is a very practical point and deserves research in traffic forecasting.

**Weaknesses:** Despite the above merits, I have to say this paper still has space to further improve.
1. Some descriptions are ambiguous. For example,  the authors claim that "Despite different semantic meanings corresponding to node-based and edge-based features, they can share some important spatial and temporal patterns." What kind of spatial and temporal patterns are not mentioned. It could be good to clarify it with a concrete example.
2. The novelty is not very high. The proposed method is based on ODE, which is a very common component. Some works have proposed similar approaches for traffic forecasting, such as [1] [2] below, which are not mentioned in the literature review. In [1], it even considers the local, global temporal embedding and edge-based dynamic embedding, which is very close to this work.
[1] MVSTT: A Multiview Spatial-Temporal Transformer Network for Traffic-Flow Forecasting
[2] Graph ODE Recurrent Neural Networks for Traffic Flow Forecasting
3. The experimental setting is not well described. How each baseline is implemented should have been suitably mentioned.
4. Analysis of the power of the proposed approach. Despite the evaluation metrics used, the authors have no analyses on the training and testing efficiency of the approach, which I think is important to learn the performance in a more comprehensive way.
5. The limitations of this work and the future work are ignored.

---

> ### Author Response · Authors · 2023-03-29
> **Response to Reviewer-MPh2**
>
> We would like to thank the reviewer MPh2 for the insightful comments. Below, we address your concerns and provide clarification on specific points.
>
> ---
> > Comment:
> > * Analysis of the power of the proposed approach. Despite the evaluation metrics used, the authors have no analyses on the training and testing efficiency of the approach, which I think is important to learn the performance in a more comprehensive way.
>
> **Response**:
> Thank you for the insightful comment. We acknowledge that our method might be computationally more expensive than the baselines. However, we would like to point out the superior performance of our multi-ODE framework, which can outweigh the increased complexity in certain use cases where prediction accuracy is of paramount importance. In response to your comment, we have taken the following steps:
>
> * To provide a clearer understanding of the performance-efficiency trade-off, we have included a performance vs. inference time pareto comparison between our method and the baselines in the Appendix (section C, Figure 7(a)).
>
> * We also present the scalability of our multi-ODE framework for larger datasets and higher-dimensional inputs in the Appendix (section C, Figure 7(b)). More discussion on the scalability of our method is included in the revised version of the paper
>
> * In the future work section of the paper, we have outlined potential strategies for enhancing our method's efficiency, providing guidance for further research in optimizing our method for real-world applications.
>
>
>
> ---
> > Comment:
> > * Some descriptions are ambiguous. For example, the authors claim that "Despite different semantic meanings corresponding to node-based and edge-based features, they can share some important spatial and temporal patterns." What kind of spatial and temporal patterns are not mentioned. It could be good to clarify it with a concrete example.
>
> **Response**:
> Thank you for pointing out the ambiguity in our sentence. In our statement, "Despite different semantic meanings corresponding to node-based and edge-based features, they can share some important spatial and temporal patterns," we are referring to the fact that both types of features can exhibit similar patterns in terms of spatial distribution and temporal evolution. For instance, consider the traffic conditions in an urban area. Node-based features, such as traffic volume at intersections, and edge-based features, such as traffic flow between intersections, may both exhibit rush hour patterns. During these periods, both the traffic volume at intersections (node-based) and the traffic flow between intersections (edge-based) could increase simultaneously, reflecting the shared temporal pattern of rush hour congestion. We will make sure that ambiguous statements are resphrased for clarity in the final version of our paper.
>
>
> ---
> > Comment:
> > * The novelty is not very high. The proposed method is based on ODE, which is a very common component. Some works have proposed similar approaches for traffic forecasting, such as [1] [2] below, which are not mentioned in the literature review. In [1], it even considers the local, global temporal embedding and edge-based dynamic embedding, which is very close to this work. [1] MVSTT: A Multiview Spatial-Temporal Transformer Network for Traffic-Flow Forecasting [2] Graph ODE Recurrent Neural Networks for Traffic Flow Forecasting
> > * The above highly related works should be added in the literature review or excluded by some reasons.
>
> **Response**:
> We appreciate your feedback and would like to point out that both [1] and [2] were released in Dec 2022 which is one month after our submission to TMLR. It is worth mentioning that our method differs from [1], which uses a spatial-temporal transformer instead of ODEs, and from [2], which employs NODE and RNNs without considering different views. We will make sure to include these references and highlight the distinctions in the final version of our paper.
>
> ---
> > Comment:
> > * The experimental setting is not well described. How each baseline is implemented should have been suitably mentioned.
>
> **Response**:
> In our study, we did not execute the baselines. We adopted their results from their original papers, using the default parameter settings provided by the respective authors. We provided further clarification on this point in the revision.

---

> > ### Author Response · Authors · 2023-03-29
> > **Response to Reviewer-MPh2 (2)**
> >
> > ---
> > > Comment:
> > > * The limitations of this work and the future work are ignored.
> >
> > **Response**:
> > We have included limitations and potential future work to the final version of paper.
> > One limitation of GRAM-ODE is its higher computational expense compared to baselines, due to its more complex model design and architecture. However, we believe that its superior performance and scalability to larger datasets may justify this complexity in scenarios where accuracy is of paramount importance. Potential future work includes investigating model compression techniques to reduce model size without sacrificing performance, exploring distributed computing strategies for efficiency, and evaluating GRAM-ODE's applicability to other spatio-temporal domains like climate modeling or social network analysis.

---

> > > ### Comment · Reviewer_MPh2 · 2023-04-08
> > > **Response**
> > >
> > > I appreciate the response from the authors, especially the additional experiments to verify the efficiency and scalability. While I'm willing to see this work is accepted, I just pose my last concern about experimental setting. That is, I doubt whether it is suitable to compare the baselines with their reported results rather than reproduce them with the same devices and environment. The comparison could be more convicing with equal repruduction.

---

> > > > ### Author Response · Authors · 2023-04-10
> > > > **Response to Reviewer-MPh2 (3)**
> > > >
> > > > We appreciate your feedback and are glad that our revisions have addressed your concerns about efficiency and scalability.
> > > > We want to point out that following prior baselines in this field, we've kept our experimental setup consistent for the fair comparison, using the same dataset splits, and the same optimization parameters, such as batch size and learning rate for each dataset. We understand that running the baselines on our devices would offer a better comparison, but conducting such reproduction for us among all seven baselines would be time-consuming for the limited response period.
> > > > We will be happy to make changes if deemed necessary.

---

### Decision · Action_Editors · 2023-04-20

**Recommendation:** Accept as is

**Comment:**

All reviewers agree that the paper is useful and the ideas are interesting. Although the idea may seem straightforward, it has never been systematically studied before, and potentially is very useful for spatio-temporal modelling of signals on the graph.

**Audience:**

1) People working with graph neural networks (looking for potential new applications).
2) People working with traffic prediction and similar applied tasks.

**Claims And Evidence:**

The paper considers deep neural network architectures for traffic modelling. It points out the disadvantages of using solely graph neural networks for this task and ignores temporal information. Instead, they propose to use combination of GNN with neural ordinary equations (NODE). The performance of the new architecture is shown on 6 real-life datasets where the improvement is shown.